# Tracking the emergence of the Upper Palaeolithic in western Asia and Europe: A Multiple Correspondence Analysis of Protoaurignacian and Southern Ahmarian lithics

Jacopo Gennai[1]*, Armando Falcucci[2]*, Vincent Niochet[3]*, Marco Peresani[4,5], Jürgen Richter[6], Marie Soressi[3]*

**1** Department of Civilisations and Forms of Knowledge, University of Pisa, Pisa, Italy, **2** Department of Geosciences, Prehistory and Archaeological Sciences Research Unit, Eberhard Karls University of Tübingen, Tübingen, Germany, **3** Faculty of Archaeology, Leiden University, Leiden, The Netherlands, **4** Department of Humanities, University of Ferrara, Ferrara, Italy, **5** Institute of Environmental Geology and Geoengineering, National Research Council, Stratigraphic, Milano, Italy, **6** Institute of Prehistoric Archaeology, University of Cologne, Cologne, Germany

* jacopo.gennai@unipi.it (JG); armando.falcucci@uni-tuebingen.de (AF); v.niochet@arch.leidenuniv.nl (VN); m.a.soressi@arch.leidenuniv.nl (MS)

## Abstract

Reconstructing changes in human behaviour during the Pleistocene, particularly when based on lithic or other artefact types, is often hindered by the traditional categorisation of these materials into discrete entities. The Early Upper Palaeolithic of Mediterranean Eurasia – comprising the Protoaurignacian, Early Aurignacian, Northern Ahmarian, and Southern Ahmarian technocomplexes – represents the first emergence of a pan-European cultural unit. However, this conventional categorisation into discrete entities obscures a deeper understanding of the dynamics of *Homo sapiens*' dispersal across Eurasia during this period. In this study, we apply Multiple Correspondence Analysis (MCA) to assess patterns of reduction processes, technological variability, and inter-assemblage homogeneity across technocomplexes. Using the comprehensive dataset provided in this paper, we analyse variability by grouping it into three domains: platform preparation, convexity management, and retouch. Solutrean Upper Palaeolithic assemblages from the Iberian Peninsula are used as an outgroup. We selected blanks, retouched and unmodified ones, and we focused on blades and bladelets, which are the typical end-product of the Upper Palaeolithic knapping. We excluded cores to avoid pitfalls of late or early reduction patterns, as our blanks cover most of the knapping sequence. We applied MCA to Early Upper Palaeolithic blanks for the first time, providing a geographically widespread comparison. Our results show that the MCA of blank attributes, particularly those describing the preparation of convexities, is sufficiently robust to reveal the distinctiveness of Early Upper Palaeolithic technologies relative to Solutrean ones. Our analysis also confirms technological similarities between the Southern Ahmarian and the

**Data availability statement:** The datasets generated and analysed in this study are available in the associated research compendium on Zenodo (https://doi.org/10.5281/zenodo.14843408). The repository includes all R scripts and derived data required to reproduce the results and figures of the study.

**Funding:** J.G.'s research on the assemblages and research at Al-Ansab 1 and Românești-952 Dumbrăvița I were funded by the German Research Foundation (Deutsche Forschungsgemeinschaft; DFG) under the umbrella of the Collaborative Research Center/SFB 806 "Our Way to Europe - Culture-Environment Interaction and Mobility in the Late Quaternary" (DFG project-code 57444011), coordinated by the University of Cologne (J.R.). Jacopo Gennai is currently supported by the "NEANDURANCE - Determining the reasons for prolonged Neanderthal survival in the Western Mediterranean area" project. Research at Fumane is coordinated by the Ferrara University (M.P.) and the project is supported by several bodies including the Italian Ministry of Culture—Veneto Archaeological Superintendence, by public institutions (the Lessinia Regional Natural Park, Fumane Municipality, BIMAdige), the Leakey Foundation (spring 2015 round), and private associations and companies. The technological analysis of the Protoaurignacian from Fumane by A.F. was supported by the German Research Foundation (Deutsche Forschungsgemeinschaft; DFG) under grant agreement no. 431809858. This project received funding from the Dutch Research Council (Nederlandse Organisatie voor Wetenschappelijk Onderzoek; NWO) 'Neanderthal Legacy' grant (VI.C.191.070) awarded to M.S. and a PhD in the humanities grant awarded to M.S. and V.N. The sponsors and funders played no role in the study design, data collection and analysis, decision to publish, or preparation of the manuscript. DFG: https://www.dfg.de/en, https://www.nwo.nl/en Dutch Research Council: https://www.nwo.nl/en.

**Competing interests:** The authors have declared that no competing interests exist.

Protoaurignacian, particularly in bladelet production, reinforcing the interpretation of bladelets as a primary production target in Early Upper Palaeolithic lithic technology. This study contributes laying the foundation for open-access databases, standardised analytical protocols, and MCA to support efforts in understanding hominin dispersal and interaction during this pivotal phase of prehistory.

## Introduction

The dispersal of *Homo sapiens* in Western Eurasia is a major anthropological topic [1–3]. The considered period features complex bio-cultural dynamics involving population and material culture replacement, which, albeit occurring almost synchronously at a large scale, also reveal regional developments [4–7]. Current theories and evidence suggest multiple scenarios [8,9], with recent research indicating at least two dispersal events of *Homo sapiens*. The first one is dated between 54–43 ka and it is now associated with the Initial Upper Palaeolithic, Bachokirian, Bohunician, Lincombian-Ranisian-Jerzmanowician, Uluzzian, and, likely, Neronian [10–19]. For some of these technocomplexes, *Homo sapiens* association is evidenced by genetic data, notably the Bachokirian and the Lincombian-Ranisian-Jerzmanowician [15,17], and others, the Uluzzian and the Neronian, by teeth morphometrical features [18,20]. In the Levant, the Levantine Initial Upper Palaeolithic is associated with a *Homo sapiens* mandible at Ksar Akil [21] and the technocomplex is linked to the Bohunician by technological resemblance [11,22]. We refer collectively to these technocomplexes as Initial Upper Paleolithic (IUP). Another potential IUP assemblage has been published from the hinterlands of Iberia dated at 44.8–42.9 ka cal BP [23]. Nonetheless, IUP technological affinities are also found in Late Mousterian assemblages in Italy [24], pointing to a more complex explanation than simple demic dispersal. Notably, genetic data from the IUP indicates little or no contribution to later Western Eurasian or modern European populations [25,26].

The second dispersal is associated with two technocomplexes: the Ahmarian and the Aurignacian [1,2,7]. However, while the European Aurignacian is associated with *Homo sapiens* genetically and morphometrically [25–28], the association between the Levantine Ahmarian and *Homo sapiens* rests upon human remains from Ksar Akil (fossil nickname Egbert) that are now lost [13,29].

Throughout the paper, we refer to the Ahmarian and the earlier facies of the Aurignacian (Protoaurignacian and Early Aurignacian) as part of the Early Upper Palaeolithic (EUP). Much of the debate on EUP technocomplexes technology focuses on laminar technology and how this enhanced *Homo sapiens* adaptability [30–33]. The EUP is roughly comprised within the 43–38 ka cal BP, after the IUP and before the advent of the Evolved Aurignacian and Levantine Aurignacian [4,34–38]. Recent research in the Levant highlights bladelets as the true game-changer [32,39], suggesting a likely discontinuity between the IUP and the EUP technologies, marked by the widespread production of bladelets believed as projectile points and part of composite tools [40–45].

Lithics are among the most commonly preserved and, consequently, frequently used proxies for human presence and behaviour in prehistoric research [46]. They provide a crucial foundation for exploring the geographic spread of similar behaviours. Yet, the traditional practice of attributing stone-tool assemblages to technocomplexes often obscures variability and limits interpretive perspectives. Shea argued for abandoning the naming of stone tool industries —the so-called NASTIES— while Reynolds and Riede compared the European Upper Palaeolithic cultural taxonomy to a "house of cards" [47,48].

Notably, the Southern Ahmarian and Protoaurignacian share techno-typological similarities [33,49]. Recent in-depth technological analysis by one of us has further confirmed a technological similarity between these traditions [50]. A recent qualitative analysis suggests that Ksar Akil layers XIII – IX, which exhibit Southern Ahmarian characteristics [51] and are dated to approximately 40 ka cal BP [13,52], align closely with the Protoaurignacian [53]. The underlying layers XIX–XVI at Ksar Akil, traditionally attributed to the Northern Ahmarian, are considered closely related to the Châtelperronian [53]. The latter interpretation needs to be carefully evaluated [54].

However, the reliance on comparing individual attributes, summarising reduction processes into broad narratives, and categorising assemblages into discrete facies or technocomplexes limits our ability to fully capture the variability of human behaviour over time and space. This variability is likely continuous, defying the rigid boundaries of these classifications [55].

In this study, we analyse the variability of four lithic assemblages attributed to the Protoaurignacian and Southern Ahmarian technocomplexes, combining technological attributes and examining their variation when grouped into technologically meaningful domains using Multiple Correspondence Analysis (MCA). MCA enables the visualization of relationships and structures among multiple categorical variables by projecting them onto a continuous, orthogonal scale [56]. This data-driven approach is inspired by [55] and [57]; it compares assemblages at the attribute level, with attributes grouped into domains [58,59], identifying similarities and patterns on a continuous scale. To ensure robust and reliable results, we focus on large lithic assemblages representing most or all reduction stages. These assemblages span the geographical breadth of the Early Upper Paleolithic (from the Levant to Western Europe) and its rough chronological range (42–38 ka cal BP). We prioritise modern excavations with sieving, which have recovered small-sized artefacts, along with reliable taphonomic reconstructions and radiometric dating. In line with the recommendations of Open Science, we openly share our data and analytical workflow to promote transparency and reproducibility [47,55,60–62]. By encouraging other researchers to combine their datasets with ours, we aim to enhance the robustness and inter-regional validity of future analyses, following the example set by Cascalheira [57,63].

## Analysis goals and tested hypotheses

We hypothesise that the EUP assemblages we examine will be more similar to one another than to any other Upper Paleolithic assemblages, assuming the distinction of the EUP is valid. To test this, we will compare our EUP assemblages to the closest-in-time and space available dataset: the one published by Cascalheira, which focuses on blade and bladelet production in Solutrean assemblages from the Last Glacial Maximum in Portugal and Spain [57,63].

Additionally, we hypothesise that assemblages classified as Protoaurignacian will be more similar to one another than to those classified as Southern Ahmarian. If this is not the case, and the two technocomplexes intermingle, it would underscore the limitations of attributing assemblages to these distinct technocomplexes. However, if Protoaurignacian assemblages are indeed more similar to one another, it could suggest an interesting geographic structuring of behaviour, as the Protoaurignacian is considered the first pan-European technocomplex, spreading from France to Bulgaria, while the Southern Ahmarian is regionally confined to the southern Levant. To test this, we will study four sites attributed to these two technocomplexes and located at significant geographic distances from one another: Al-Ansab 1 in Jordan, Românești-Dumbrăvița I in Romania, Grotta di Fumane in Italy, and Les Cottés in France.

We also aim to test whether the 12 mm width threshold used to distinguish bladelets from blades holds across all assemblages. This threshold was originally introduced by Tixier [64] to systematise the Maghreb Epipalaeolithic. During the last years, it has been accepted as the most common dimensional threshold to distinguish between blades and bladelets within the EUP [50,65–67].Ultimately, we seek to evaluate whether the methodology used here can help better understand the homogeneity and regional variability of the Early Upper Palaeolithic.

## The different facies of the Early Upper Paleolithic

EUP assemblages are described from sites spanning the Levant, the Caucasus, the southern East European Plain, and most of Europe (Fig 1). The study and the definition of EUP technocomplexes have a long history of research ([68]). We also provide a comprehensive list of EUP sites considered for the distribution map and a list of radiometric dates obtained with modern methods (S2 and S3 Files).

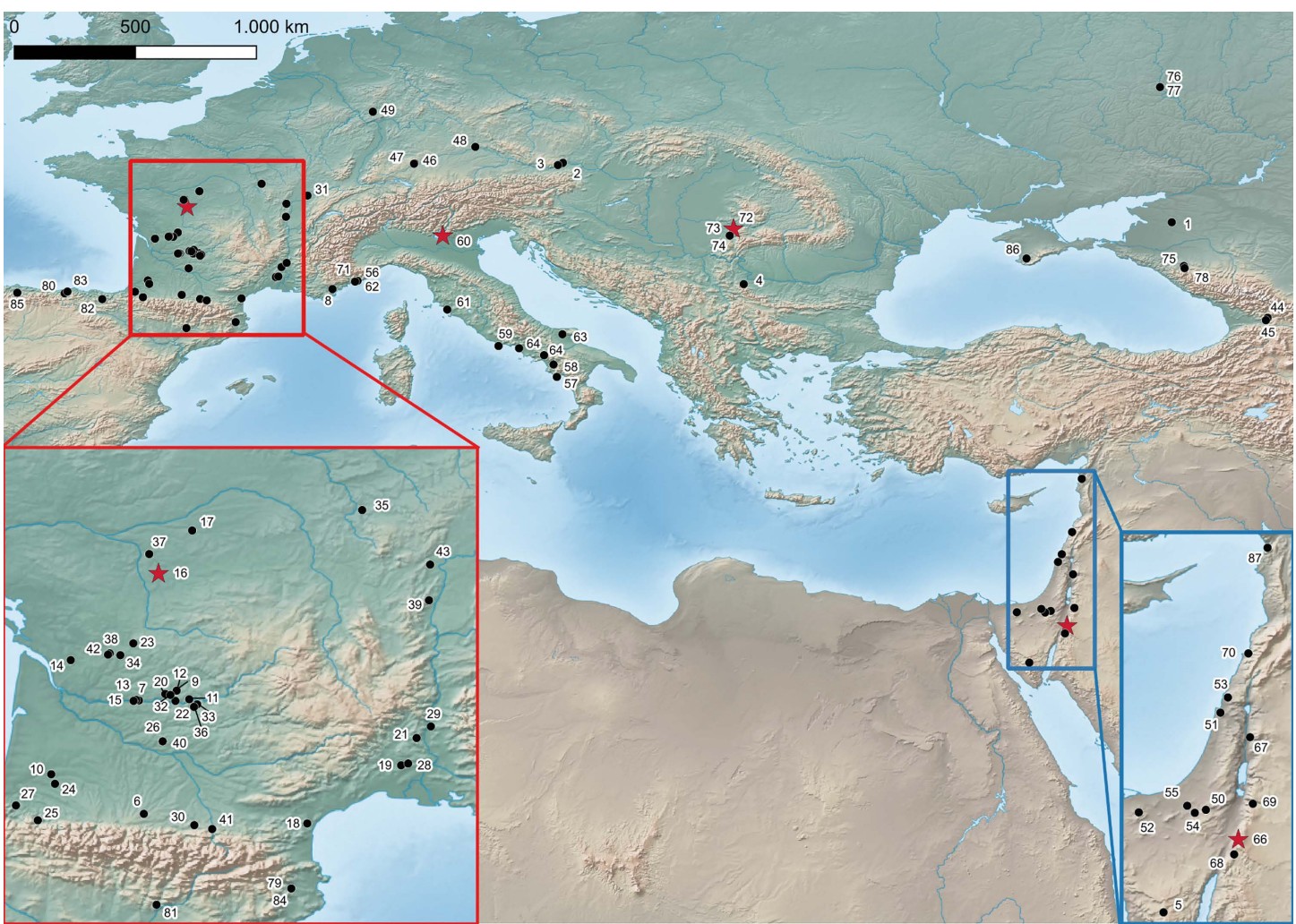

**Fig 1. EUP sites.** The stars are corresponding to the sites analysed: 66 – al Ansab 1, 73 – Românești Dumbrăvița I, 60 – Grotta di Fumane, 16 – Les Cottés. The rest of the sites can be accessed in the S3 File.

The Ahmarian is divided into two main *facies* based on technological and metrical features: the Northern Ahmarian and the Southern Ahmarian [49]. The two *facies* occupy distinct geographical and environmental areas and do not appear within the same stratigraphic sequence [69]. In the Northern *facies*, there are mostly blade cores exploited with a bidirectional pattern, while in the Southern one cores are exploited with a unidirectional pattern. The Northern *facies* focuses on blades, with rare unidirectional bladelet cores, while the Southern one integrates blade-bladelets in the same reduction or produces just bladelets [70–73]. The Aurignacian features an internal variability, that is intensively debated. For the past two decades, the earliest Aurignacian is often portrayed as split into two *facies* or technocomplexes: the Protoaurignacian and the Early Aurignacian. The two *facies* are variously interpreted as chronological phases, as the Protoaurignacian always occurs first in stratigraphical sequences, or adaptations to different ecological niches, being the Protoaurignacian often circum-Mediterranean distributed, while the Early Aurignacian occurs further North or in colder climates [31,74]. This picture is valid for most of Western Europe [42,66,75], while elsewhere in Europe discrepancies emerge [67]. The Protoaurignacian lithic technology features a continuous reduction of pyramidal, convergent edges, volumetric cores to produce small and slender blades and large and slender bladelets, the latter compose the bulk of retouched implements transformed in variously marginally retouched bladelets and most noticeably the Dufour bladelet sub-type Dufour [31,42,66,76–78]. The Early Aurignacian lithic technology features a disjointed production of larger blades from prismatic, parallel edges, volumetric cores and small bladelets from carinated cores, in this case, bladelets are rarely retouched [31,66,76,78–80].

Radiocarbon dating features a prime spot in the debate and narrative of EUP hominins dispersals and technocomplexes filiation. Despite the Protoaurignacian being generally older and always beneath the Early Aurignacian in stratigraphical sequences, there is a degree of overlapping in the first occurrences of both facies at the continental level [35,74,81,82]. Current radiometric dating is too coarse to assess infra-millennial developments around 40 ka cal BP. The Southern Ahmarian looks younger than the Protoaurignacian [68], but we need to consider the considerable efforts in modern radiometrically dating the European contexts, that produced older dates [83,84]. Additionally, the dating of Ahmarian contexts suggests a potential overlap between the two *facies* [10,13,52,69,85,86]. However, the absence of both facies within the same stratigraphic sequence prevents the determination of their chronological relationship in terms of anteriority and posteriority. The anteriority of the Northern Ahmarian relies on the dates obtained at Manot cave, Kebara cave, and the set of dates obtained at Ksar Akil by Bosch and colleagues [34,52,87]. These determinations are either showing a large timespan (Manot cave – [34]) or are disputed by other authors [88,89]

## Materials and methods

We will study three assemblages attributed to the Protoaurignacian, excavated at the sites of Românești-Dumbrăvița I in Romania, Grotta di Fumane in Italy, and Les Cottés in France, along with one assemblage attributed to the Southern Ahmarian from Al-Ansab 1 in Jordan. These assemblages have been excavated using modern methods that ensure the recovery of small-sized artefacts, analysed for taphonomy and post-depositional processes, and dated using radiometric techniques.

All the necessary permits required for the study were obtained from the institutions and privates listed in the acknowledgements section.

Below we will present the sites and the previous studies and interpretations of the assemblages we will study here.

### Presentation of the studied sites

**Al-Ansab 1.**  Al-Ansab 1 (hereafter Ansab) is located in the Lower Wadi Sabra (30°14′2.4′′N 35°22′58.8′′E; 618 m above sea level) [69]. Archaeological excavation ran from 2009 to 2020. The archaeological artefacts are embedded in sands and gravels originating from fluvial and aeolian deposits. The site is an open-air location, and the preservation of archaeological features and archaeological artefacts is unaffected by significant post-depositional processes, especially

in the northern area of the site [90]. Charcoal recovered in AH1 shows that the site was occupied during a brief span between 38–37 ka cal BP [69,90]. The site is excavated using a 1-m$^2$ basic grid unit, which is further subdivided into quadrants of 0.25 m$^2$. Layers are geological and are excavated in arbitrary 5-cm-deep spits. Finds ≥ 10 mm in maximum dimension have been individually piece-plotted using a total station since 2015. Finds > 20 mm have two or more points plotted to record the contour. Smaller finds are identified by quadrant and spit number alongside finds retrieved by dry sieving through a 2-mm mesh.

**Românești-Dumbrăvița I.** Românești-Dumbrăvița I (hereafter Românești) is located on a river terrace overlooking the confluence of the Bega Mare and Bega Mica rivers near the Românești village, in Timiș county, Western Romania portion of the Banat (45°49′2.45″ N, 22°19′15.85″ E, 212 m above sea level) [41]. The location is open-air with two archaeological *loci* Românești I and II, lying 80 m apart: Românești I is by far the most extensive [41,91]. Faunal and organic remains, in general, are extremely rare due to the preservation conditions [41]. The first investigations at the site and digging of large portions of the area happened between during the 1960's and the early 1970's [41,91]. A new testpit occurred at the margin of the older trenches in 2009 and it was expanded in 2016, 2018, and 2019 (Chu et al. 2022; Sitlivy et al. 2012). All investigations provided a largely similar stratigraphical sequence featuring the top soil, a layer with Epigravettian lithics (GH 2), a layer with Aurignacian lithics (GH3), and a final layer with few flakes signalling an earlier occupation before the Aurignacian [41]. Optically stimulated luminescence (OSL) and thermoluminescence (TL) dates bracketed the Aurignacian artefacts to between 42.1 and 39.1 ka, with a mean age of 40.5 ka [92]. The 1 m$^2$ basic grid unit is further subdivided into quadrants of 0.25 m$^2$, and digging in these later excavations has proceeded in 2 cm deep spits confined within each geological horizon [41]. Finds over 5 mm are spatially recorded using a total station, sediments were wet sieved with a 5mm mesh and selected quadrants with a 2 mm mesh [41].

**Grotta di Fumane.** Fumane is located in the western Monti Lessini Plateau within the Venetian Prealps of northeastern Italy [93]. The site has been continuously excavated since 1982, Fumane is a cave site and it contains a long stratigraphic sequence, spanning from MIS 4 to the Heinrich Event 3, when the cave ceiling eventually collapsed [93]. Macro-unit A includes multiple layers attributed to the Mousterian, Uluzzian, Protoaurignacian, Early Aurignacian, and Early Gravettian [93]. The Protoaurignacian layer A2-A1 date to around 42 and 40 ka cal BP [84,94]. and represent some of the oldest Aurignacian assemblages associated with *Homo sapiens* remains ([28]. These layers predate Heinrich Event 4, as confirmed through the small-mammal assemblage analysis [95]. Zooarchaeological data suggest that the site was occupied seasonally during late spring and summer, with a focus on exploiting ibex and chamois [94]. Additionally, the data point to a cold environment, characterised by mostly open landscapes and patchy woodlands. The Protoaurignacian layers are rich in anthropogenic content, with clear combustion features, dumps, and occupation horizons [96,97]. In addition to the abundant lithic industries, the site is renowned for the discovery of a large marine shell assemblage, indicating the use of ornamental objects sourced at least 400 km away [98]. A2-A1 was excavated using a stratigraphic method, with all artefacts larger than 1.5 cm recorded within their respective sub-square meters of provenience (33x33 cm). Both dry and wet sieving of excavated sediments were systematically conducted to recover the smallest organic and inorganic artefacts.

**Les Cottés.** The site of Les Cottés (46°41′44″N 0°50′40″E; 90 m above sea level) is located in the Poitou region in central-western France, at the northern limit of the Aquitaine Basin, between the cities of Poitiers and Tours, in the village of Saint-Pierre-de-Maillé [66]. The cave opens in a Jurassic limestone cliff, about 30 m high, which dominates the Gartempe river, located nowadays about 150 m to the East. Known since late 19$^{th}$ century, the site has been the object of several excavation campaigns. The interior of the cave was excavated in 1880–1881 [99–101]. Then the platform at the entrance of the cave was excavated two times in the second half of the 20$^{th}$ century [102–105]. Between 2006 and 2018, M. Soressi led an update of the stratigraphic and chrono-cultural context based on the sections left by the previous excavators, as well as an extension of the excavated surface [66]. This recent excavation resulted in significant advances in radiometric dating [106,107], archaeozoology [108], lithic technology [42,109,110], palaeoenvironmental reconstructions

[111] and aDNA analysis [112,113]. A total of 15 m$^2$ disposed in a U-shape in front of the cave were excavated. Sediment accumulation primarily results from colluvial deposits from the plateau above and erosion of the cliff. All the layers exhibit a regular slope descending towards the South-East. The stratigraphy consists of nine units, six of which contain archaeological assemblages, spanning from at least 43.1 ka cal BP for the Mousterian assemblage (US 08) to 36,400 cal BP for the uppermost Late Aurignacian assemblage (US 02) ([107], calibration on OxCal4.4 using IntCal20). Single quartz grain OSL and MET-pIRIR dating place the US 08 at 51±3 ka and US02 at 37.2±1.5 ka [106]. The Protoaurignacian assemblage found in US 04 inférieure is radiocarbon-dated to 40.1–38.9 ka cal BP ([107], calibration on OxCal4.4 using IntCal20). The single quartz grain OSL date of US 04 inférieure, 41±2 ka, is comparable to the radiocarbon one [106]. Archaeozoological data show a progressive evolution of the environmental conditions from a steppic to an arctic landscape [111,114]. In US 04 inférieure, the disappearance of temperate species indicates colder conditions than those of lower assemblages. The upper part of US 04 (referred to as supérieure), attributed to the Early Aurignacian, is often separated from US 04 inférieure by a thin sterile layer. US 04 inférieure is often separated from the underlying US 06 (attributed to the Châtelperronian) by a 15 cm-thick low-density layer, US 05. A total of 5,351 pieces greater than 1,5 cm were analysed. Raw materials mostly come from local sources, while about 20% of the pieces come from the Grand Pressigny region (20–40 km to the North) and 5% from more distant areas (over 40 km from the site: [66,115]).

## Previous studies and interpretations of the studied assemblages

The assemblages have been the object of previous independent studies.

**Al-Ansab AH 1.** The assemblage has been studied to retrieve technological behaviours and mobility assessment. The first analyses by Hussain and Parow-Souchon [72,116] provided the attribution to the Southern Ahmarian technocomplex. The analysed assemblage consisted of the artefacts retrieved during the 2009–2013 excavations campaign that mostly interested an erosional step. Parow-Souchon interprets the assemblage as the result of multiple residential mobility occupations that left a wide range of lithics and a complete reduction sequence due to the undifferentiated activities on site and vicinity of the raw material sources. In 2018 the analysis resumed by one us (J.G.) to contextualise the bladelet production and provide a continental comparison of the EUP technologies. In addition to the 2009–2011 coordinated artefacts, the analysed sample comprised coordinated artefacts from squares excavated in the 2018 campaign (Fig 2). Technology at Al-Ansab AH 1 involved a repetitive and standardised scheme. Raw material nodules feature an oblong shape; therefore, the flaking surface is generally placed on the shorter face and reduction progresses frontally. Striking platforms are plain and the knapping angles are very acute, resulting in strong distal convexity. The start of the lamino-lamellar reduction is often placed around natural lateral ridges, which then merge into one single flaking surface. Very few formal bifacial crests are present. At the time of discard, cores show a semicircumferential shape, or they retain the narrow-faced shape. Knapping products are mainly blades and bladelets, as flakes intervene mostly during the earliest phases of the core roughing out and during part of the core flank management. Some of these flakes are then recycled in burin cores. Gennai's interpretation primarily differs from that of Parow-Souchon and Hussain regarding the role of bladelets in the reduction process. While Parow-Souchon and Hussain predominantly interpreted the assemblage as blade-oriented [116], Gennai considered the abundance of bladelets and their role within the reduction process as evidence that they were the primary focus of production, with blades representing only a minor component. Bladelets-sized negatives are often found on the flat part of the flaking surfaces and encased by lateral blade-sized negatives. Bladelets-sized negatives are often found intercalated with blades and on blades dorsal faces [50,71].

**Românești-Dumbrăvița I GH3.** The artefacts from 2016–2019 with single coordinates have been fully analysed by one of us (J.G.) to provide a technological and taxonomic assessment [41,50]. The analysis showed that a complete reduction process is present on-site, mostly using locally sourced raw materials. The production is focused on the obtention of bladelets from volumetric, unidirectional cores (Fig 3). Cores are either semicircumferential or narrow-faced. Despite the assemblage being blade-bladelet oriented, there is a significant amount of non-cortical flakes. The main interpretation is

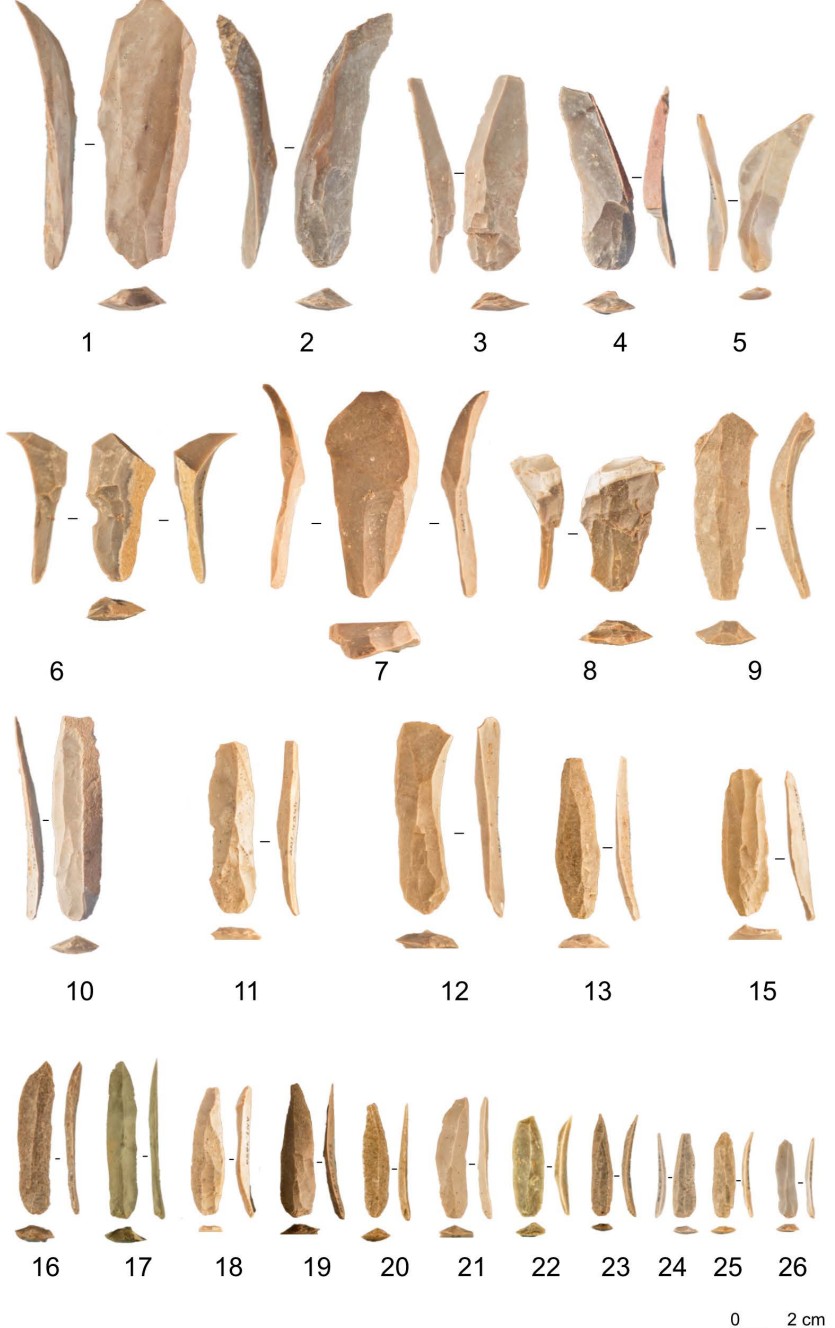

**Fig 2. Sample of al-Ansab 1 AH1 blades and bladelets included in the analysis.** 1–5 asymmetrical blades, 6–9 overshot blades, 10–15 simple blades, 16–26 simple bladelets. Pictures Jacopo Gennai.

that the lower quality of the used raw material influenced the knappers' core preparation and that flakes might come from various core management activities, such as partial striking platform rejuvenation. Bladelets are mostly produced from the central flat part of the flaking surface and are generally encased by blade-sized negatives [50]. The assemblage has been attributed on techno-typological and dating grounds to the Protoaurignacian [41].

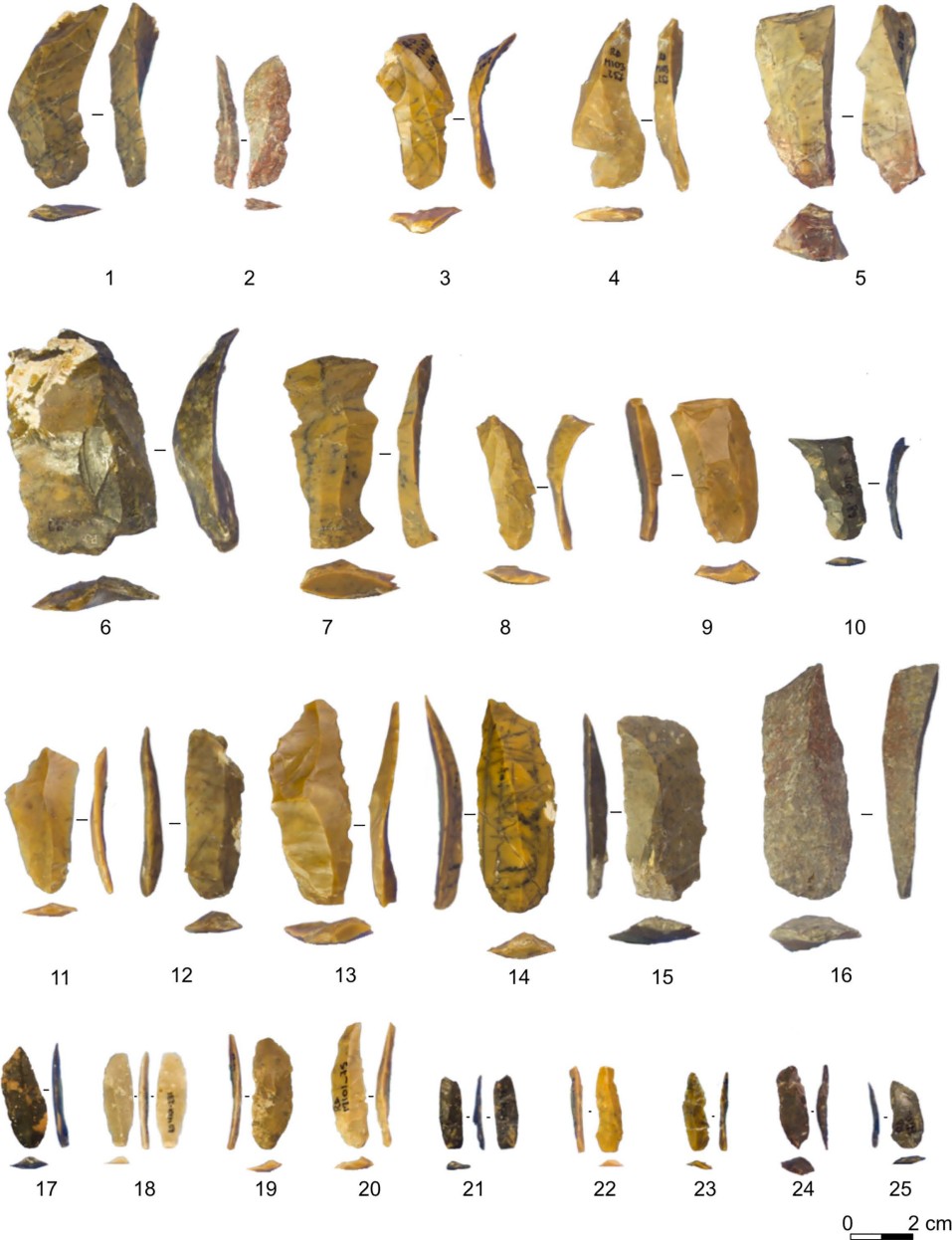

**Fig 3. Sample of Românești-Dumbrăvița I GH3 blades and bladelets included in the analysis.** 1–5 asymmetrical blades, 6–10 overshot blades, 11–16 simple blades, 17–25 simple bladelets. Pictures Jacopo Gennai.

**Grotta di Fumane A2-A1.** Fumane is one of the key sites in Mediterranean Europe for understanding the technological and behavioural variability of the Protoaurignacian. As such, its lithic assemblages have been analysed over the years by several scholars [50,65,117–120]. In addition to traditional technological approaches, the earliest Protoaurignacian assemblages have been studied using functional [121], 3D geometric morphometrics, and reduction intensity approaches. Recently, Falcucci and colleagues [75] assessed the integrity of the Aurignacian lithic assemblages using a break connection method [122] to conjoin broken blades, further combining spatial analysis and lithic taphonomy.

Their study showed that A2 and A1 should be considered a single analytical unit, characterised by palimpsest formation and marked spatial variability. Therefore, in this study, the two assemblages are merged and analysed together as A2-A1. Regarding the spatial sample, the lithics studied in this paper come from the cave exterior and the area around the drip line, where postdepositional processes are less pronounced compared to the cave interior [75,120]. At Fumane, complete reduction sequences were carried out on-site, with evidence of core initialisation, maintenance, and retooling activities. Bladelet production was mostly based on the use of platform unidirectional cores, with marginal percussion used to extract slender bladelets. In most cases, striking platforms are plain, and reduction procedures were aimed at isolating convergent flaking surfaces to extract pointed and relatively straight bladelets, which were frequently modified by marginal retouching (Fig 4). Carinated technology was used only marginally to produce short and curved bladelets. The dataset analysed in this study is a subset of the main Fumane dataset published on Zenodo [123] and associated with the recent reanalysis of the Aurignacian deposit. The Zenodo dataset contains all Aurignacian and Gravettian lithics from the entire excavation area.

**Les Cottés US04-inf.** The lithic assemblage analysed here (Fig 5) was retrieved from the lower part of the stratigraphic unit 04 (US 04 inférieure) attributed to the Protoaurignacian [66]. The typological spectrum is largely dominated by retouched bladelets, followed by marginally retouched blades, which outnumber scrapers, burins and other tools. This proportion confirms the differentiation from Early Aurignacian contexts, such as US 04 supérieure [42,66]. The débitage mainly aims at the production of bladelets, using primarily unidirectional reduction processes. Bladelets are produced either through a somewhat flexible frontal reduction modality on narrow surfaces of varied flint volumes, or through a more standardised convergent reduction modality on wide surfaces, requiring the removal of convergent elongated products from the sides of the flaking surface. This behaviour has recently been emphasized in many Protoaurignacian contexts (e.g., [50,65]). The production of blades stems either from a semicircumferential modality, or sometimes from a frontal modality on narrow surfaces.

## Control group

To refine our understanding of the lithic reduction attributes in our dataset, we incorporated a control group consisting of blanks attributed to the Solutrean period dated to the Last Glacial Maximum and excavated in Spain and Portugal. This dataset compiled by Cascalheira [57,63] is one of the few freely available and reusable datasets, and, most importantly, it is comparable with our EUP assemblages. The Solutrean technology relies on volumetric reduction for producing blades and bladelets, like the EUP, but is chronologically distinct enough – circa 20,000 years – to exhibit its unique lithic reduction signature. Including the Solutrean control group serves multiple purposes. First, it provides a comparative benchmark against which the patterns in our EUP dataset can be assessed. Despite the technological similarities, the Solutrean data's distinct chronological position may reveal unique characteristics and variations in lithic reduction practices. This comparison helps validate the clusters and patterns identified in our analysis, strengthening the reliability of our findings. Cascalheira's dataset, derived from extensive technological analyses of Iberian Solutrean assemblages, offers a detailed and open-access record of lithic attributes. This dataset is particularly valuable because it includes artifact-level entries rather than just frequency or presence/absence data, which is rare in open-access technological datasets. We performed attribute homogenisation to ensure meaningful integration of this control group with our dataset. This process aligned the attributes from both datasets to minimise biases and ensure comparability, allowing us to incorporate the Solutrean data effectively into our analysis.

## Composition of the database

We focused on complete blanks and mediodistal or medioproximal fragments that preserve a significant portion of the original blank. We use the term blank as an equivalent of debitage product [124,125]. This approach allows for the inclusion of attributes relevant to specific parts of the blank—for instance, platform attributes apply only to blanks with a

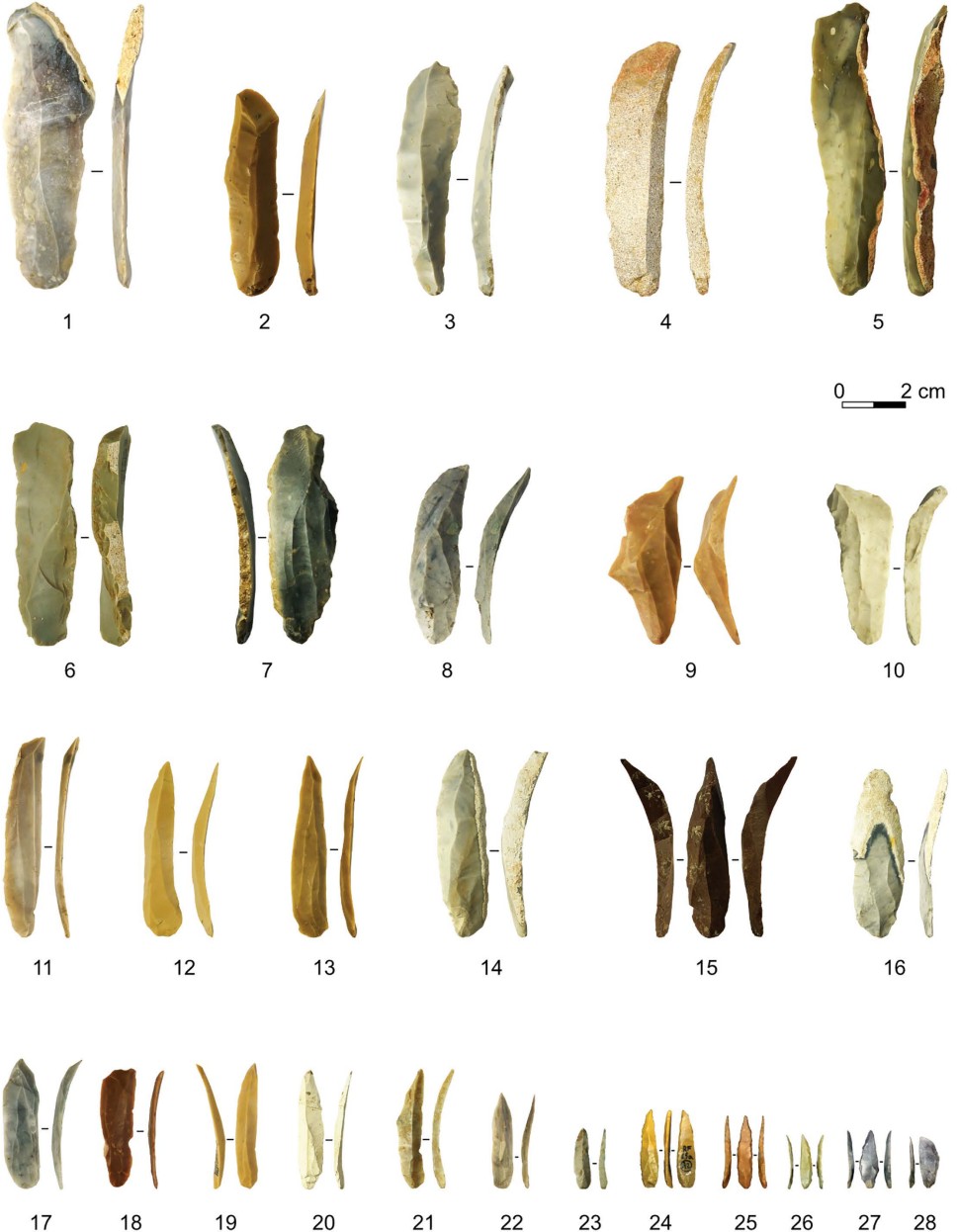

**Fig 4. Sample of blades and bladelets from Fumane A2-A1 included in the analysis.** 1, 5, 14 Semi-cortical blades, 2–4 Simple blades, 6 Neo-crested blade, 7, 16 Semi-cortical blades with bladelet removals, 8–10, 15 Lateral blades, 11–13, 19–23 Simple bladelets, 14 Naturally backed semi-cortical blade, 17–18 Small blades with bladelet scars, 24–28 Bladelets with lateral retouch. Photos: Armando Falcucci.

preserved platform (i.e., the proximal part), while attributes such as distal end morphology are assessed only in artefacts retaining a distal portion. At the same time, it maintains qualitative rigour by excluding smaller fragments, such as isolated distal, medial, and distal fragments, which may lead to erroneous observations due to their highly localised characteristics.

Reproducibility is a delicate matter in lithic studies, and it has a severe impact on the understanding of prehistoric human behaviours, as lithics are one of the most common sources of information for the Palaeolithic. Nonetheless, few studies delved

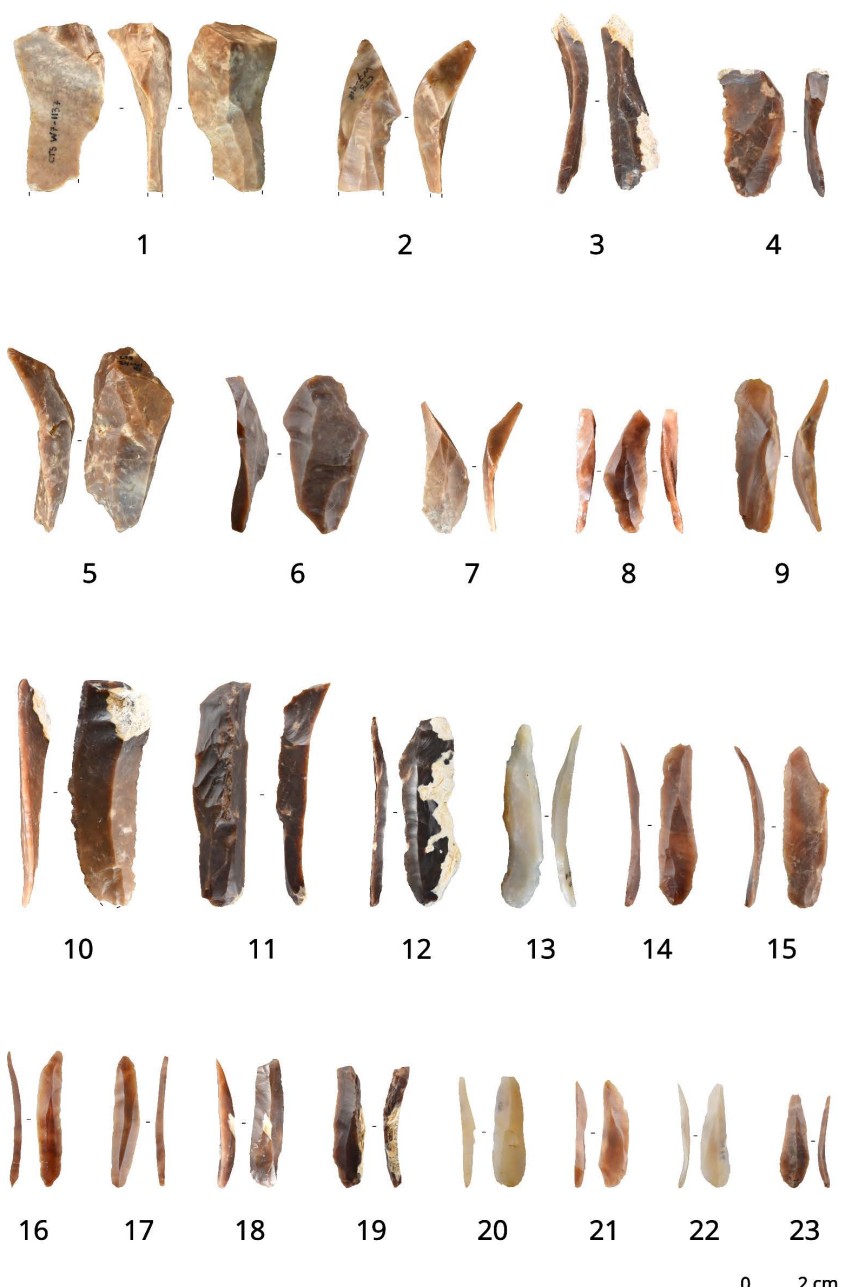

**Fig 5. Sample of Les Cottés US 04inf blades and bladelets included in the analysis.** 1–4 overshot blades, 5–8 asymmetrical blades, 9–15 simple blades, 16–23 simple bladelets. Pictures Leonardi Carmignani (4, 6, 9, 11, 13-17, 19-23) and Vincent Niochet (1-3, 5, 7-8, 10, 12, 18).

into the inter-analysts' reproducibility and the problem of reproducibility impacts more qualitative analysis approaches than quantitative ones. The four assemblages analysed in this study were examined separately by analysts trained in the *chaîne opératoire* approach. They collected both qualitative and quantitative attributes (Table 1), with the latter known for its higher reproducibility index [126]. The data collection did not follow a controlled experiment, but they were collected using traditional standard attribute definitions [124,125,127]. Even though we used similar attributes and definitions, adjustments were needed.

**Table 1. Attributes and their homogenisation process. Equivalent attributes in the databases of each author, new attribute names in the merged database, and the meaning of the attribute.**

| Gennai Attribute Name | Niochet Attribute Name | Falcucci Attribute Names | New Names | Explanation | List of variable within the attribute | Used in domain |
|---|---|---|---|---|---|---|
| Site | Site | Site | Site | assemblage site | | |
| | Technocomplex | | Technocomplex | Technocomplex | | |
| Layer | Layer | Layer | Layer | stratigraphical unit | | |
| Piece Original ID | Piece Original ID | ID | ID | ID of the single artefact | | |
| Entirety | Entirety | Preservation | Preservation | whether the artefact is complete or fragmentary. If fragmentary Proximal, Medio-Proximal, Medial, Medio-Distal, Distal | "Complete" "Medioproximal" "Mediodistal" | |
| | | Cortex.y.n | Cortex.y.n | if there is cortex or not | "Yes" "No" | |
| CxSimpl | CxSimpl | Cortex | Cortex | Cortical surface extent | "No Cortex" "Semi" "Extensive" "Full" | |
| CxPosition | CxPosition | Cortex.position | Cortex.position | position of the cortical surface on the artefact | NA "Lateral" "Distal" "Dorsal" "Proximal+Distal" "Proximal" "Proximal+lateral" "Dorsal+lateral" "Distal+lateral" "Distal+proximal" "Dorsal+distal"ND" | |
| Length | Length | Length | Length | artefact length on its technological axis | | |
| Width | Width | Width | Width | artefact width at artefact mid-length perpendicular to the technological axis | | |
| Thick | Thick | Thickness | Thickness | artefact thickness at artefact mid-length perpendicular to the width | | |
| EI | EI | Elongation | Elongation | artefact length/artefact width | | |
| | | robustness | Robustness | Artefact width/artefact thickness | "high" "medium" "low" | Platform |
| ButtType | ButtType | platform.type | Platform | type of platform | "Linear" "Plain" NA "Punctiform" "ND" "Cortical" "Facetted" | Platform |
| BulbMorph | BulbMorph | bulb.type | Bulb | type of Bulb | "Marked" "Diffuse" "NA" "ND" | |
| Lipp | Lipp | Lip | Lip | presence of a lip, recorded on laminar artefacts | "Yes" "No" | |

*(Continued)*

| Gennai Attribute Name | Niochet Attribute Name | Falcucci Attribute Names | New Names | Explanation | List of variable within the attribute | Used in domain |
|---|---|---|---|---|---|---|
| OvAb | OvAb | dorsal.thinning | Abrasion | presence of overhang abrasion on the proximal dorsal surface | "Yes" "No" | |
| Axiality | Axiality | axiality | Axiality | if the artefact technological axisi correspond or not to its morphological axis | "Yes" "No" | Convexity |
| Out | Out | blank.shape | Outline. morphology | dorsal view of the artefact | "Parallel" NA "Convergent" "Off-axis" "ND" | Convexity |
| Symmetry | Symmetry | cross.section. symmetry | Symmetry | whether the artefact cross section is symmetric or not at mid-length | NA "asymmetric" "symmetric" "ND" | Convexity |
| CrossSect-Morph | CrossSectMorph | cross.section | Cross.section | artefact cross section shape at mid-length | "ND" "Polyhedric" "Triangular" NA | Convexity |
| Pro | Pro | curvature | Profile | artefact longitudinal profile | "Straight" "Twisted" "Curved" ["Slightly Curved" "ND" NA | Convexity |
| | | Torsion | Torsion | whether the artefact's longitudinal profile is twisted or not | "Yes" "No" | Convexity |
| DEndMorph | DEndMorph | distal.end.profile | Distal.end.morpho | artefact's termination longitudinal profile | "Feathered" NA "ND" "Plunging" | Convexity |
| | | | Blank.type1 | Flake or Laminar | | |
| StructureCat | StructureCat | blank | Blank.type2 | Flake, Blade, Bladelet | | |
| TechCat | TechCat | technology | Technology.Phase | phase of the reduction | "Ordinary" "MGMT" "Init" | |
| Cat | Cat | technology.ext | Technology.Ext | technological category | "Ordinary bladelet" "Ordinary blade" "Lateral asymmetrical laminar" "Overshot laminar" "Flaking surface maintenance laminar" "Crested laminar" "Maintenance laminar" Core tablet" Burin spall" "Cortical flake" "Ordinary flake" "Flaking surface maintenance flake" "Maintenance flake" | |

*(Continued)*

| Gennai Attribute Name | Niochet Attribute Name | Falcucci Attribute Names | New Names | Explanation | List of variable within the attribute | Used in domain |
|---|---|---|---|---|---|---|
| NegN | NegN | scar.count | Number.negatives | number of negatives on the dorsal surface | | |
| NegType | NegType | bladelet.neg.blade & blade.neg.flake | Negatives.type | | "Bladelets" "ND" "No negatives" "Flakes" "Blades" "Blades & Bladelets" "Blades & Flakes" "Bladelets & Flakes" NA | |
| | | | Dorsal.scar.1 | Negatives orientation along the technological axis, simplified | "Unidirectional" NA "Crossed" "Unidirectional+Other direction" "Bidirectional" "Orthogonal" "Bidirectional+Other direction" "ND" "Centripetal" "Other" | |
| NegO | NegO | scar.pattern | Dorsal.scar.2 | Negatives orientation along the technological axis | "Unidirectional" "Unidirectional+Convergent" NA "Crossed" "Unidirectional+Orthogonal" "Bidirectional" "Convergent" "Orthogonal" "Unidirectional+Crossed" "Convergent+Orthogonal" "Crossed+Orthogonal" "Unidirectional+Convergent+Orthogonal" "Bidirectional+Crossed" "Bidirectional+Convergent" "Unidirectional+Crossed+Orthogonal" "ND" "Bidirectional+Orthogonal" "Unidirectional+Convergent+Crossed" "Unidirectional parallel" "Unidirectional convergent" "Unidirectional transverse" "Centripedal" "Other" "Multidirectional" "Undetermined" "Crested" "Unidirectional + Orthogonal" "Convergent + Bidirectionnal" "Cortical" "Convergent + Bipolar" "Convergent + Orthogonal" | |

(Continued)

**Table 1.** (Continued)

| Gennai Attribute Name | Niochet Attribute Name | Falcucci Attribute Names | New Names | Explanation | List of variable within the attribute | Used in domain |
|---|---|---|---|---|---|---|
| R | R | Class | Tool | whether the artefact is retouched or not | "Yes" "No" | |
| Rpos | Rpos | retouch.position | Retouch.Position | Retouch position on the artefact's faces | NA "_" "Alternate" "Direct" "Inverse" | Retouch |
| Rloc | Rloc | RLoc | Retouch Location | retouch localisation on the artefact | NA "_" Bilateral "Right" "Distal" "Proximal" "Left" "Right Proximal" "Distal+Mesial" "Left Distal" "Lateral" "Distal+Right" "NA" "Proximal+Left" "Distal+Left" "Proximal+Right" "Undetermined" "Distal+Proximal" "Right + Mesial" | Retouch |
| Rdis | Rdis | RDist | Retouch Distribution | retouch extent on the artefact | NA "_" "Continuous" "Partial" "Undetermined" "Continuous + Discontinuous" | Retouch |
| Typology | Typology | typology | Typology | Synthetic tool type determination | | |

The result is a database comprising 6698 entries across the four assemblages. AN accounts for 2050 entries, ROM for 1094 entries, FUM for 2715 entries, and CTS04inf for 839 entries (Table 6).We defined blades, bladelets and flakes according to standard criteria: a blade and a bladelet feature subparallel lateral edges and an elongation of 2 or greater, with a metrical threshold of 12 mm in width separating blades from bladelets [64,124,125]. A unimodal histogram of blade and bladelet width is typically interpreted as evidence of continuous knapping, with a gradual transition from blades to bladelets [57]. To assess the universality of this metrical threshold, we plotted the distribution of blade and bladelet widths using 1 mm bins. We ensured comparability by analysing similarly sized samples, excluding retouched blanks, and adjusting the sample size to match the smallest assemblage (510 blanks from Românești-Dumbrăvița I GH3). For the assemblages from Al-Ansab 1 AH1, Grotta di Fumane A2-A1, and Les Cottés 04-inf, sampling was conducted while maintaining the original proportions of blades and bladelets. The width values of the sampled artefacts were grouped into 1 mm intervals, and our analysis compared the median and mode widths of these samples against a threshold of 12 mm, which is commonly used to differentiate between blades and bladelets.

**Al-Ansab 1 AH1.** The Al-Ansab 1 AH1 (AN) database consists of 2050 entries, corresponding to 948 blades, 809 bladelets and 293 flakes (Table 2). The technological analysis study sample consists of single plotted complete and semi-complete blanks and cores recovered during the 2009–2011 and 2018 campaigns. The sample is a casual one encompassing areas with the highest concentrations of artefacts. Flakes tend to be complete, while blades and bladelets are fragmentary at least by half.

The AN assemblage consists of mainly high-quality and local tabular cherts found in nearby outcrops (<1 km) [116]. The whole lithic reduction is found on-site and no difference is noted between the different raw materials [50,116].

**Românești-Dumbrăvița I GH3.** The Românești-Dumbrăvița I GH3 (ROM) database consists of 1094 entries, corresponding to 262 blades, 288 bladelets and 544 flakes (Table 3). The sample consists of the whole piece-plotted complete and semi-complete artefacts excavated in 2016–2019, excluding square P104. Flakes tend to be complete, while blades and bladelets are fragmentary at least by half.

The ROM assemblage shows mostly local (<10 km) procurement (Ciornei et al., 2020). Blocs were found in primary, sub-primary locations or river gravels and imported on-site as minimally modified cores [128]. Longer-distance raw materials (13–60 km) were imported as prepared cores too [128]. A single artefact is made of Carpathic obsidian and imported as a finished tool [41,128]. Therefore, most of the reduction process happened on-site and no difference in the lithic

**Table 2. Composition of Al-Ansab 1 AH1 (AN) sample.**

| | Complete | | Mediodistal | | Medioproximal | | Total N | Total % |
|---|---|---|---|---|---|---|---|---|
| | N | % | N | % | N | % | | |
| Blade | 531 | 56.01% | 238 | 25.11% | 179 | 18.88% | 948 | 100.00% |
| Bladelet | 311 | 38.44% | 195 | 24.10% | 303 | 37.45% | 809 | 100.00% |
| Flake | 269 | 91.81% | 11 | 3.75% | 13 | 4.44% | 293 | 100.00% |
| | 1111 | 54.20% | 444 | 21.66% | 495 | 24.15% | 2050 | 100.00% |

**Table 3. Composition of Românești-Dumbrăvița I GH3 (ROM) sample.**

| | Complete | | Mediodistal | | Medioproximal | | Total N | Total % |
|---|---|---|---|---|---|---|---|---|
| | N | % | N | % | N | % | | |
| Blade | 131 | 50.00% | 48 | 18.32% | 83 | 31.68% | 262 | 100.00% |
| Bladelet | 106 | 36.81% | 95 | 32.99% | 87 | 30.21% | 288 | 100.00% |
| Flake | 481 | 88.42% | 18 | 3.31% | 45 | 8.27% | 544 | 100.00% |
| | 718 | 65.63% | 161 | 14.72% | 215 | 19.65% | 1094 | 100.00% |

reduction process is noticed between the different raw materials [41]. The local raw material is often described as of lower knapping quality, featuring internal cracks and a coarser texture, nevertheless, it did not impede the technological goals and the development of a frankly Aurignacian assemblage [41].

**Grotta di Fumane A2-A1.** The Grotta di Fumane A2-A1 (FUM) database consists of 4647 entries, corresponding to 1065 blades, 2996 bladelets, 581 flakes, and 5 undetermined (Table 4). Finds bigger than 15 mm were coordinated during excavation, the analysis focused on these artefacts. A third of the blades and flakes are complete, while only around 15% of the bladelets are complete. 40% of the bladelets are medial fragments. FUM is characterised by abundant retouched blanks, especially bladelets, which may have skewed fragment representation. The dataset is a subset of the main Fumane dataset, which is available under a CC BY 4.0 license on Zenodo [123].

The FUM assemblage consists mostly of high-quality flint embedded in the carbonate formations of the western Monti Lessini, ranging from the Upper Jurassic to the Middle Eocene. They are available within 5–15 km from the site. The most common, determined with macroscopic features, are cherts embedded in the Maiolica, the Scaglia Rossa, the Scaglia var-iegata, and the Ooliti di San Virgilio formations [65]. Flint also abounds in loose coarse streams or fluvial gravels, slope-waste deposits, and soils in the immediate surroundings of the cave [129]. Jurassic and Tertiary calcarenites, frequently found in large-sized and homogeneous nodules, were almost exclusively used to produce blades [117].

**Les Cottés US04-inf.** The Les Cottés US04-inf (CTS04inf) database consists of 839 entries, corresponding to 476 blades, 353 bladelets and 10 flakes (Table 5). Our original selection consisted of 1303 complete and sub-complete blades, bladelets and informative flakes. However, only two-thirds (64,5%) of them were made on local raw materials (a local lacustrine flint mostly and some upper Turonian marine flint). As only local materials were used in the three other sites studied here, we consequently chose to excluded the artefacts on raw materials coming from more than 20 km of the site and the ones which remained of undetermined origins. This led to reduce the potential technological and statistical biases that would have stemmed from different economic patterns between sites.

Finds bigger than 15 mm were coordinated during excavation. A few pieces were retrieved in the sieves and allocated an individual identification. About 20% of blades and bladelets and a third of flakes are complete. More than half of each category are medioproximal fragments. Around one-quarter of blades and bladelets are mediodistal fragments. Artefacts

**Table 4. Composition of Grotta di Fumane A2-A1 (FUM) sample. The list excludes three angular debris listed in the dataset by [123], as they cannot be associated with any specific blank class.**

| | Almost complete | | Complete | | Proximal | | Medioprox-imal | | Medial | | Mediodistal | | Distal | | Undeter-mined | | Total N | Total % |
|---|---|---|---|---|---|---|---|---|---|---|---|---|---|---|---|---|---|---|---|
| | N | % | N | % | N | % | N | % | N | % | N | % | N | % | N | % | | |
| Blade | 19 | 1.80% | 287 | 27.13% | 71 | 6.62% | 303 | 28.17% | 251 | 23.72% | 119 | 11.15% | 15 | 1.42% | | 0.00% | 1065 | 100.00% |
| Bladelet | 15 | 0.47% | 472 | 15.79% | 89 | 3.01% | 862 | 28.81% | 1170 | 39.06% | 352 | 11.73% | 36 | 1.12% | | 0.00% | 2996 | 100.00% |
| Flake | 9 | 1.56% | 214 | 36.85% | 81 | 14.01% | 210 | 36.16% | 19 | 3.11% | 36 | 6.23% | 6 | 1.04% | 6 | 1.04% | 581 | 100.00% |
| Undetermined | | 0.00% | | 0.00% | | 0.00% | | 0.00% | 2 | 40.00% | | 0.00% | | 0.00% | 3 | 60.00% | 5 | 100.00% |
| | 43 | 0.91% | 974 | 21.04% | 241 | 5.22% | 1375 | 29.54% | 1442 | 30.99% | 507 | 10.89% | 57 | 1.17% | 11 | 0.24% | 4647 | 100.00% |

**Table 5. Composition of Les Cottés US04-inf (CTS04inf) sample after selecting only local raw materials.**

| | Complete | | Medioproximal | | Mediodistal | | Total N | Total % |
|---|---|---|---|---|---|---|---|---|
| | N | % | N | % | N | % | | |
| Blade | 103 | 21.64% | 249 | 52.31% | 124 | 26.05% | 476 | 100.00% |
| Bladelet | 74 | 20.96% | 195 | 55.24% | 84 | 23.80% | 353 | 100.00% |
| Flake | 3 | 30.00% | 6 | 60.00% | 1 | 10.00% | 10 | 100.00% |
| | 180 | 21.45% | 450 | 53.64% | 209 | 24.91% | 839 | 100.00% |

come mainly from the northern and eastern areas of the recently excavated surface, where each stratigraphic unit is well-separated by low-density layers and post-depositional processes are minimal.

## Merging the sites' databases into one

The database was produced using a similar analytical approach and employed interoperable terms. Nevertheless, some observations required homogenisation—at the very least in terms of formatting, capitalisation, and terminology—to ensure proper processing in R [130]. The software R was chosen to handle all processes of data wrangling and analysis to foster reproducibility due to its open-source nature and widespread adoption in data analysis [130]. The homogenisation process, resulted in a merged database containing 37 attributes, most of which were already present in the original databases and have been renamed, while others were derived from existing data (Table 5) using the functions available in the R Tidyverse environment [131]. The code used for data manipulation and attribute homogenisation and analysis is provided as SI file and available on Zenodo and Github alongside all datasets (see Data Availability statement).
Changes included:

- Preservation: almost complete blanks from Fumane have been registered as complete ones.

- Cortex: originally the AN, ROM, and CTS04inf databases showed cortex presence in 25% steps. The FUM database in 33% steps. After carefully reviewing occurrences in blanks and their technological role we decided to rename them as semicortical blanks with up to 50% (AN, ROM, CTS04inf) and up to 66% (FUM) cortical surface. Blanks above these thresholds are renamed Extensively cortical. Blanks having 0% cortex have been renamed to No cortex, those with 100% cortex are fully cortical.

- Cortex position: the position of the cortex on the blanks' dorsal faces featured too many observations, some being single observations. This would have hindered the comparability. Therefore, the cortex position observations have been changed accordingly to distal, distal and lateral, distal and proximal, dorsal, dorsal and distal, dorsal and lateral, lateral, proximal, proximal and lateral, and undetermined. Blanks without cortex presence have been left blank.

Table 6. Final merged database composition.

| | Complete | | Mediodistal | | Medioproximal | | N total | % total |
|---|---|---|---|---|---|---|---|---|
| | N | % | N | % | N | % | | |
| **AN** | **1111** | **36.75%** | **444** | **38.79%** | **495** | **19.53%** | **2050** | **30.58%** |
| Blade | 531 | 17.59% | 238 | 20.84% | 179 | 7.06% | 948 | 14.16% |
| Bladelet | 311 | 10.24% | 195 | 16.99% | 303 | 11.95% | 809 | 12.04% |
| Flake | 269 | 8.91% | 11 | 0.96% | 13 | 0.51% | 293 | 4.38% |
| **CTS04inf** | **180** | **5.96%** | **209** | **18.30%** | **450** | **17.75%** | **839** | **12.53%** |
| Blade | 103 | 3.41% | 124 | 10.86% | 249 | 9.82% | 476 | 7.11% |
| Bladelet | 74 | 2.45% | 84 | 7.36% | 195 | 7.69% | 353 | 5.27% |
| Flake | 3 | 0.10% | 1 | 0.09% | 6 | 0.24% | 10 | 0.15% |
| **FUM** | **1011** | **33.50%** | **329** | **28.81%** | **1375** | **54.24%** | **2715** | **40.55%** |
| Blade | 306 | 10.14% | 119 | 10.42% | 303 | 11.95% | 728 | 10.87% |
| Bladelet | 487 | 16.14% | 174 | 15.24% | 862 | 34.00% | 1523 | 22.75% |
| Flake | 218 | 7.22% | 36 | 3.15% | 210 | 8.28% | 464 | 6.93% |
| **ROM** | **718** | **23.79%** | **161** | **14.10%** | **215** | **8.48%** | **1094** | **16.34%** |
| Blade | 131 | 4.34% | 48 | 4.20% | 83 | 3.27% | 262 | 3.91% |
| Bladelet | 106 | 3.51% | 95 | 8.32% | 87 | 3.43% | 288 | 4.30% |
| Flake | 481 | 15.94% | 18 | 1.58% | 45 | 1.78% | 544 | 8.13% |
| **Total** | **3020** | **100.00%** | **1143** | **100.00%** | **2535** | **100.00%** | **6698** | **100.00%** |

- Platform: platform types have been reduced to cortical, plain, linear, punctiform, facetted, and undetermined. We merged dihedral and facetted platforms in a more general facetted variable, as the most common platform difference within the EUP is plain or facetted, therefore we equated the platforms with more than one scar as facetted [31]. Blanks without a proximal part, i.e., mediodistal ones, have been left blank. Concave (AN, ROM, CTS04inf) and double (FUM) platforms have joined plain ones. Dihedral platforms have joined facetted ones. Natural platforms have been named cortical. Crushed platforms in the AN and ROM databases joined the undetermined ones, while in CTS04inf they joined the linear ones after the observer noticed they mostly related to this category. Abraded platforms (FUM) joined the undetermined ones.

- Outline morphology: the dorsal shape view's observations have been reduced to convergent, parallel, off-axis. Only CTS04inf kept the "other" observation. The platform attribute has been left blank in case of flake, tablet or medioproximal blank.

- Cross section: the shape of the transversal cross-section has been reduced to polyhedric or triangular. Polyhedric has been preferred to trapezoidal. This attribute has been left blank in case of flake or tablet blanks.

- Profile: The artefact longitudinal profile observations have been reduced to straight, slightly curved, curved, and twisted. Twisted was not present in FUM, as it is expressed by a separate attribute: Torsion. Therefore, AN and ROM blanks with a calculated curvature value and a twisted profile could be changed into straight, slightly curved, or curved. Those that did not have a calculated value or CTS04inf blanks kept the twisted observation. Whether a longitudinal profile is twisted or not is expressed by the new attribute

- Torsion: This attribute has been left blank in case of flake or tablet blank.

- Distal end morphology: the artefact's termination longitudinal profile has been reduced either to feathered or plunging. Hinged terminations have been joined to the undetermined, and the stepped terminations have been left blank: both these observations did not give any technological information.

- Dorsal scar 1: was derived and rationalised from dorsal scar 2. Observations were reduced to unidirectional, bidirectional, centripetal, crossed, orthogonal, other, and undetermined. In case another direction was joining the unidirectional or bidirectional variant, but they were not prevalent, the observation unidirectional/bidirectional+other direction was used.

## Variance analysis

We conduct a detailed variance analysis of attributes observed on different types of blanks—flake, blade, and bladelet. Initially, we explore the frequencies of these attributes within each blank class, comparing them between sites using the R packages ggstatsplot [132] and ggplot2 [133] for calculation and visualisation.

We then analysed sets of attributes by grouping them into technologically meaningful [58] domains. A similar approach is described in [59] and in [57]. Our Convexity and Platforms domains have strong similarities with those defined by [58] (specifically, the Dorsal Surface convexity domain and the Platform maintenance domain) and grouped the variables into three domains:

- **Platform domain**: This domain includes the Platform and Abrasion attributes and examines their relationship with the Robustness index (width divided by thickness). We hypothesise that less-prepared platforms are associated with the absence of abrasion and blanks with low robustness index—i.e., those with a smaller ratio between width and thickness, indicating they are relatively thicker or more compact in cross-section. Also, we hypothesise that given the shape, blanks with a high robustness index would result in wider platform types like linear or plain. The Platform domain groups the

Platform and Abrasion attributes and tests them against Robustness. Blanks with undetermined values, those that do not preserve the proximal part, cortical platforms, and flakes are excluded. The Solutrean dataset lacks the Abrasion attribute.

• **Convexity Domain**: This domain encompasses attributes such as Axiality, Outline, Symmetry, Cross-section shape, Torsion, Profile, and Distal end longitudinal profile. These attributes help define the products of the reduction process and infer their role and position within the core reduction. We assume that skewed, bent, and irregular shapes indicate management products—typically involving the removal of lateral and distal core ends to create convexities—while on-axis, straight, and regular shapes correspond to target products, which do not primarily aim to produce convexities. Blanks with undetermined or missing values, as well as flakes, are excluded from the analysis. The Solutrean dataset lacks an attribute reporting cross-section symmetry, and observations like "Divergent" and "Biconvex" outlines, not being recorded in the other assemblages, were removed.

• **Retouch Domain**: This domain groups the attributes of Retouch position, location, and distribution [125].

To visualise and analyse the associations within these domains, we used Multiple Correspondence Analysis (MCA – [56]). MCA was performed using the FactoMineR package [134], and the results were plotted along two most significant orthogonal axes of variation. Attribute observations are then positioned within a two-dimensional space, forming clusters that are colour-coded based on their contribution to explaining variance.

We began the analysis by organising categorical variables into a Burt table—a contingency table that displays the frequency of each category and their co-occurrences. The diagonal blocks of the Burt table show single variable frequencies (e.g., the number of Plain platforms), while the off-diagonal blocks show co-occurrences (e.g., the number of Plain platforms with Abrasion). We apply Singular Value Decomposition (SVD) to the Burt table to extract principal components, representing directions in which the data varies the most. The analysis focuses on a bidimensional representation by selecting the first two principal components. To enhance interpretation, we used supplementary variables (also called passive or illustrative variables) in the MCA plot. These supplementary variables, while not included in the initial principal components calculations, are projected into the same factor space to provide context for the clusters without altering the structure defined by the active variables. Supplementary variables in this analysis include blank type – blade and bladelet -, the name of the assemblages studied, and the technocomplexes to which they are attributed. The labels for the supplementary variables in the plots are as follow:

**AN.bladlt** = Al-Ansab AH1 bladelets

**AN.Blade** = Al-Ansab AH1 blades

**ROM.bladlt** = Românești-Dumbrăvița I GH3 bladelets

**ROM.Blade** = Românești-Dumbrăvița I GH3 blades

**FUM.bladlt** = Grotta di Fumane A1-A2 bladelets

**FUM.Blade** = Grotta di Fumane A1-A2 blades

**CTS04inf.bladlt** = Les Cottés 04inf bladelets

**CTS04inf.Blade** = Les Cottés 04inf blades

The supplementary categories representing technocomplexes are:

**Solu** = Solutrean

**Proto** = Protoaurignacian

**S.Ahm** = Southern Ahmarian

To further understand the clusters formed by the attributes and supplementary categories, we computed distance matrices using the 'factoextra' package [135]. These matrices are visualised through heatmaps, where colours range from red (indicating strong association or no distance) to blue (indicating weak association or maximum distance). This visualisation helps identify closely related attributes, though it does not define precise mid-distance score cutoffs. The distance matrices were calculated using the Euclidean distance between the active variables and the supplementary categories on the biplot (first two dimensions), assessing similarity based on the coordinates derived from the MCA. We used the `get_dist` function from the 'factoextra' package for this calculation.

## Results

### Exploratory plots

We compare frequencies of attributes' observations across blanks, assemblages, and technocomplexes [68]. The results highlight similarities between the different assemblages. Most blanks are non-cortical, particularly bladelets (SIFig 4 in S1 Fig). Laminar blanks most commonly exhibit lateral and distal cortical positions, while flakes tend to have a higher proportion of dorsal cortex (SIFig. 6 in S1 Fig). Platforms are predominantly non-facetted (plain, linear, or punctiform – SIFig 8 in S1 Fig). Abrasion of the proximal part is frequently observed in both blades and bladelets (SIFig. 11 in S1 Fig). Blades and bladelets typically have a regular, on-axis shape (SIFig. 12 in S1 Fig), with bladelets tending to be more convergent in silhouette and triangular in cross-section (SIFIg. 14, SIFig. 17 in S1 Fig). A straight or slightly curved profile is the norm for laminar artefacts, whereas curved profiles are more common in blades (SIFig. 19 in S1 Fig). Plunging distal ends are more frequently seen in blades, though they are not predominant (SIFig 21 in S1 Fig). Twisted artefacts are rare (SIFig. 23 in S1 Fig). Unidirectional knapping direction is overwhelmingly present in both blades and bladelets (SIFig. 25 in S1 Fig). Retouch positions show distinct patterns across sites: while blades are predominantly retouched on their dorsal face, bladelets exhibit a progressive increase in retouch on the ventral face as one moves westward (from AN to CTS04inf – SIFig. 27 in S1 Fig).

### Metrical data of blades and bladelets from the four tested assemblages

We present here the histograms of blades and bladelets' widths for each assemblage. We use width values, as length measurements are more influenced by the shape and dimensions of the flaking surface. The histograms for each assemblage show that the median width of the blanks is close to the 12 mm threshold. Notably, the peaks of the histograms occur around or below this threshold. Specifically, the mode, which represents the bin with the highest frequency of observations, consistently falls below the 12 mm threshold across all assemblages. For instance, the mode values are between 9–10 mm for AN, 11–12 mm for ROM, 9–10 mm for FUM, and 10–11 mm for CTS04inf. The ROM data show two prominent peaks: one between 8–9 mm (41 counts) and another slightly higher between 11–12 mm (42 counts), indicating two dominant size clusters. In contrast, the other sites—AN, FUM, and CTS04inf—display unimodal distributions. Furthermore, the median widths of the combined blades and bladelets' samples are 12.3 mm for AN, 11.8 mm for ROM, 10.9 mm for FUM, and 13.3 mm for CTS04inf. This suggests that while there is a range of blade and bladelet types within the samples, bladelets are particularly well-represented, especially in the FUM assemblage (Fig 6). We tested the same excluding mediodistal fragments, while the mode values remain similar in the four assemblages, the combined blades and bladelets median value increase in AN and ROM towards 12.5 mm (SIFig. 40 in S1 Fig). We also compared the width values of Protoaurignacian, Southern Ahmarian and Solutrean assemblages, without finding visbile variations between them (SIFig. 41 in S1 Fig).

### Multiple Correspondence Analysis

**Platform domain.** The Platform domain, with the Solutrean assemblages alongside Early Upper Paleolithic (EUP) assemblages, consists of 6183 blade and bladelet artefacts. The first two dimensions of the correspondence analysis

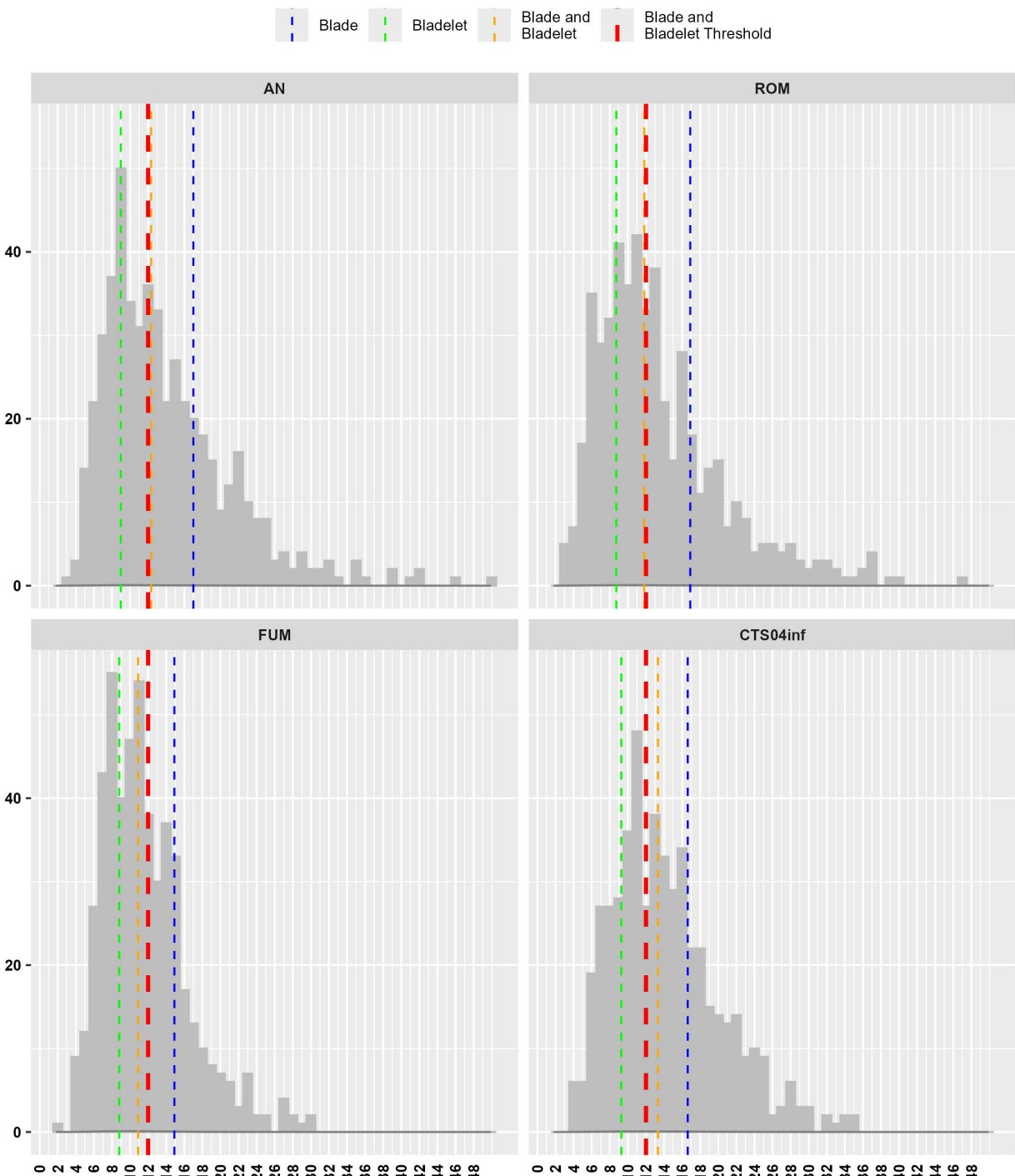

**Fig 6. Histogram of blades and bladelets width for each of the studied samples, values are binned by 1 mm.** Blades and bladelets are lumped together. Dashed lines represent median values (blue = blades, green = bladelets, orange = all blanks) and the arbitrary 12 mm threshold between blades and bladelets (red). AN is corresponding to Al-Ansab 1, ROM to Românești-Dumbrăvița I GH3, FUM is Fumane A2-A1, CTS04inf to Les Cottès 04 inférieure.

explain only 38.8% of the total variance, with Dimension 1 accounting for 21.0% and Dimension 2 for 17.8% (Fig 7). Dimension 1 highlights a contrast between Cortical platforms and High robustness (blanks that are relatively thinner). Low robustness (i.e., blanks relatively thicker) is primarily associated with Cortical platforms and the Solutrean technocomplex, as confirmed by their closer Euclidean distances. Conversely, Linear platforms are linked with High robustness, reflecting their shorter Euclidean distance to this attribute. The Protoaurignacian and Southern Ahmarian technocomplexes cluster closer to punctiform platforms and are strongly associated with slender blanks. In contrast, the Solutrean technocomplex is predominantly linked to thicker blanks relative to their width. Overall, the Protoaurignacian and Southern Ahmarian assemblages, cluster on the left side of Dimension 1, further illustrating their association with Punctiform platforms and slender blanks, in contrast to the Solutrean's association with Cortical platforms and thicker blanks.

Focusing on the EUP assemblages, 3929 blade and bladelet artifacts were included in the Platform domain analysis. The first two dimensions of the correspondence analysis explain 40.0% of the total variance, with Dimension 1 accounting for 25.1% and Dimension 2 for 14.9% (Fig 8). Dimension 1 is primarily defined by the contrast between the presence and absence of abrasion, with the absence of abrasion strongly associated with cortical platforms. In contrast, the presence of abrasion is closely linked to FUM blades and bladelets, as well as CTS04inf bladelets. Dimension 2 highlights the opposition between linear and plain platforms. High Robustness is strongly associated with linear platforms, while medium Robustness (9.07–17.8) is more closely linked to plain platforms. Based on their positions in the biplots and their Euclidean distances, CTS04inf blades show a stronger association with plain platforms, while AN and ROM bladelets are more closely related to Linear platforms. Overall, all sites cluster closely along Dimension 1, reflecting shared characteristics, but are distributed along Dimension 2 in a pattern corresponding to their geographic distances.

**Convexity domain.** A total of 3834 blades and bladelets were included in the comparison between the Solutrean assemblages and the EUP assemblages for the convexity domain. The first two dimensions explain 62.1% of the total variance, with Dimension 1 accounting for 45.2% and Dimension 2 for 16.9% (Fig 9). Dimension 1 highlights the contrast between off-axis and on-axis morphologies. Off-axis attributes, along with the presence of torsion and plunging distal terminations, form one cluster, as indicated by their Euclidean distances. In contrast, axial morphologies cluster at the opposite end, characterised by the absence of torsion, symmetric cross-section shapes, and convergent outlines. The Protoaurignacian and Southern Ahmarian technocomplexes trend toward the off-axis cluster, although their distances place them closer to polyhedric cross-sections. The Solutrean technocomplex clusters at the opposite pole, with AN blades associated with plunging and slightly curved profiles, while ROM bladelets, AN bladelets, and CTS04inf blades form another distinct cluster. FUM blades and ROM blades are associated with polyhedric shapes and curved profiles, while FUM bladelets mostly cluster with the absence of torsion and regular morphologies, distinct from Solutrean blades and bladelets, which cluster around convergent and feathered profiles. Dimension 2 is characterized by twisted profiles, plunging distal end morphology, the presence of torsion, and curved profiles. Active variables associated with Dimension 2 include twisted and curved profiles, as well as feathered and plunging distal end morphologies. The supplementary categories most strongly associated with Dimension 2 are AN blades, CTS04inf bladelets, and FUM blades. Dimension 2 reveals an opposition between two clusters: one formed by twisted profiles and the presence of torsion, and the other by curved profiles and plunging distal terminations. No technocomplex correlates clearly with these clusters, although the Southern Ahmarian is closest to the curved profile cluster. Overall, the Solutrean dataset clusters internally with regular morphologies, the Protoaurignacian positions centrally, and the Southern Ahmarian trends toward the opposite end of the Solutrean. The Protoaurignacian and Southern Ahmarian are more closely related to each other than either is to the Solutrean.

Focusing on the EUP assemblages, 2192 blades and bladelets were included in the analysis. The first two dimensions account for 62.7% of the total variance, with Dimension 1 explaining 45.4% and Dimension 2 contributing 17.3% (Fig 10). Dimension 1 emphasizes the contrast between off-axis and on-axis morphologies. Off-axis

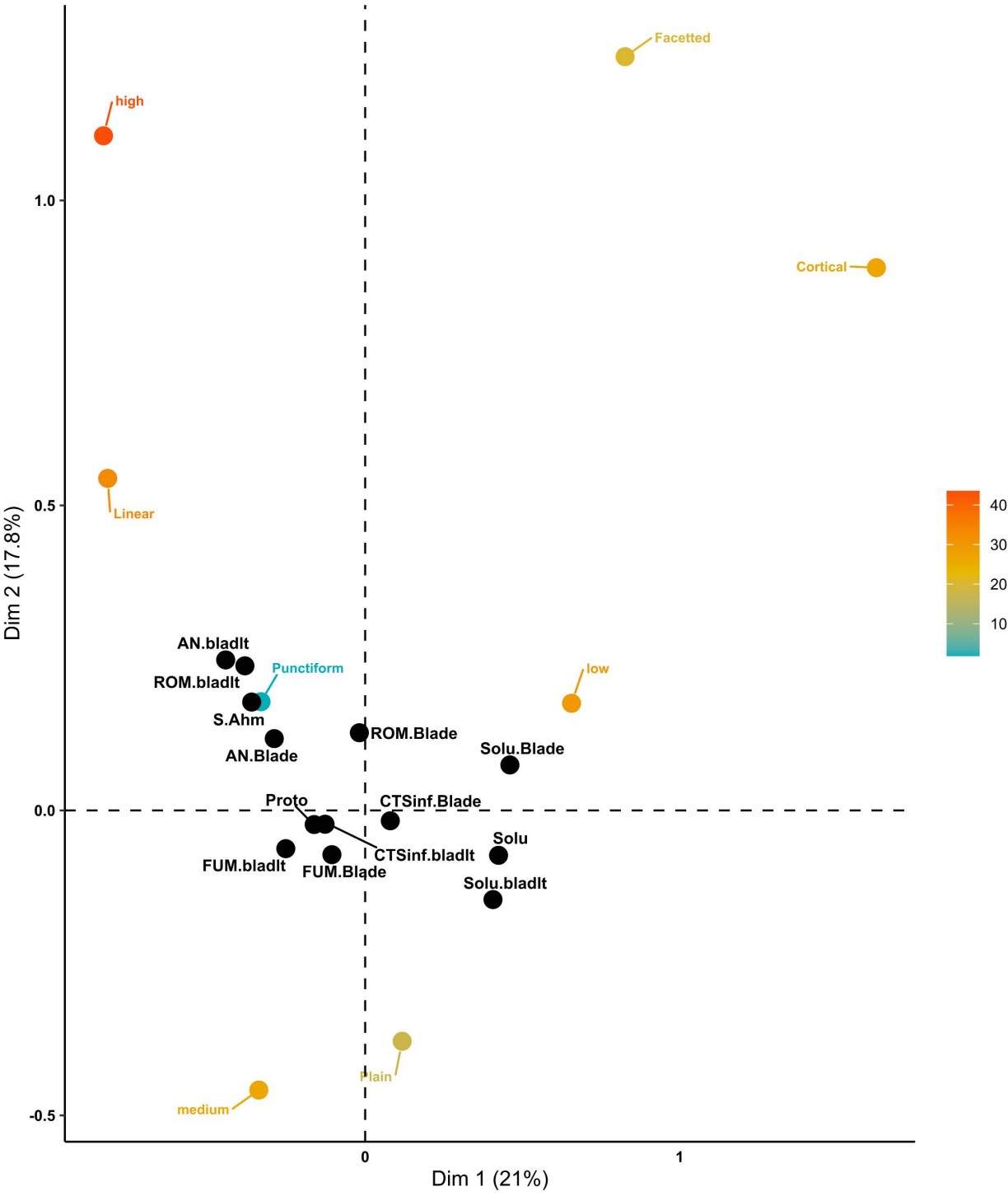

**Fig 7. Platform domain MCA biplot of EUP and Solutrean assemblages.** The active categoriesare coloured from light blue to dark orange according to their total contribution to the two dimensions. The supplementary qualitative categories are coloured in black.

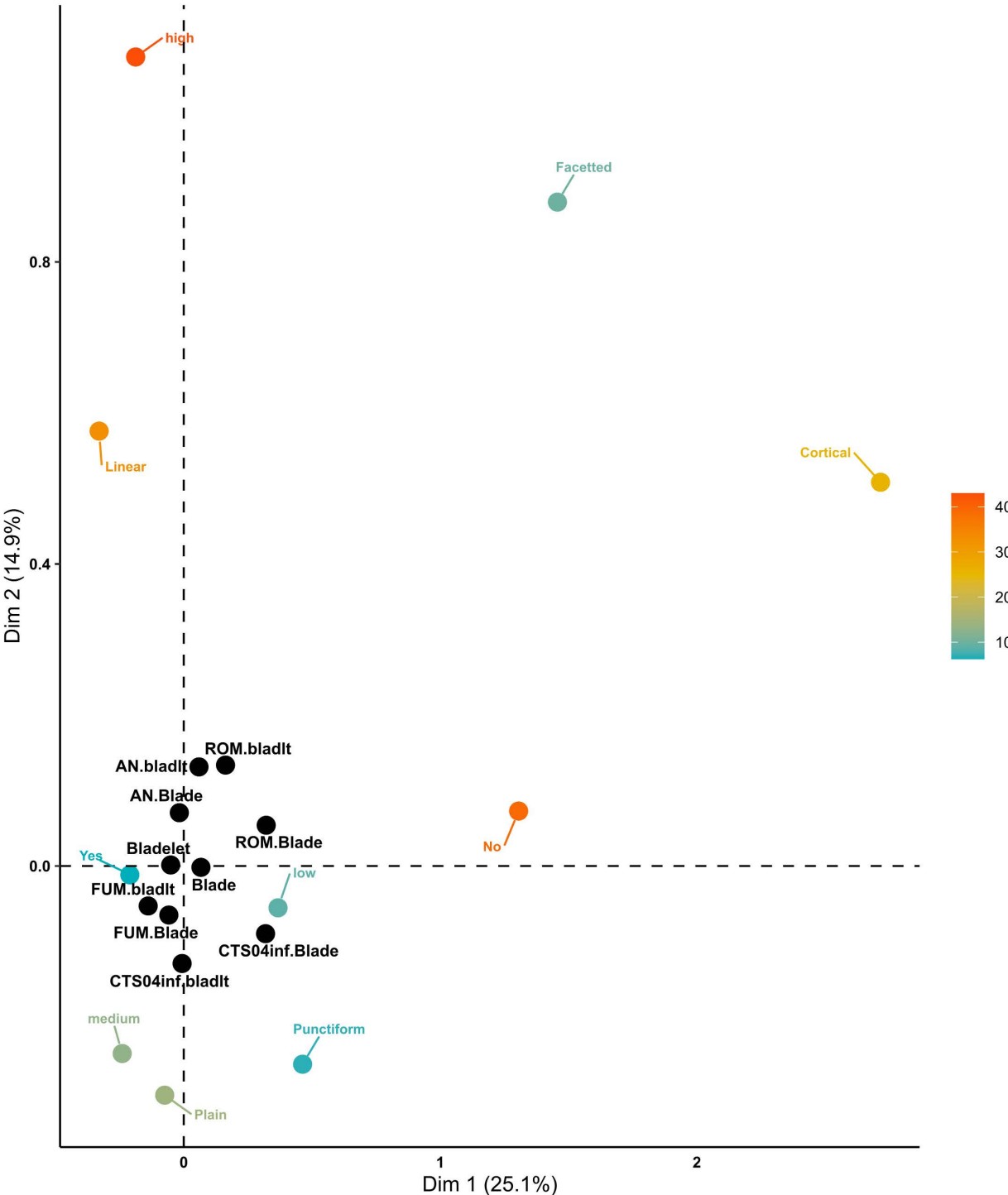

**Fig 8. Platform domain MCA biplot of EUP assemblages.** The active categoriesare coloured from light blue to dark orange according to their total contribution to the two dimensions. The supplementary qualitative categories are coloured in black.

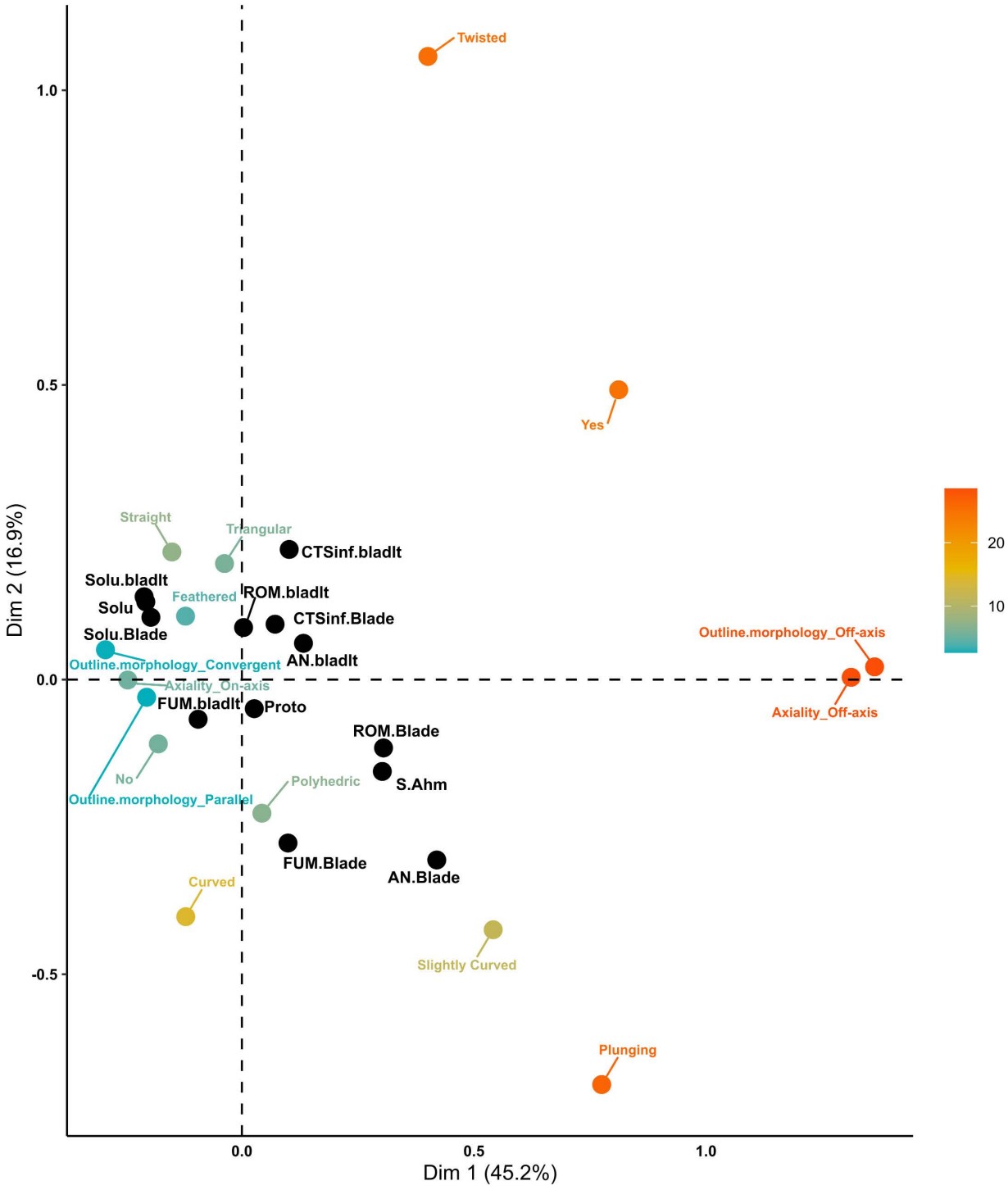

**Fig 9. Convexity domain MCA biplot of EUP and Solutrean assemblages.** The active categories are coloured from light blue to dark orange according to their total contribution to the two dimensions. The supplementary qualitative categories are coloured in black.

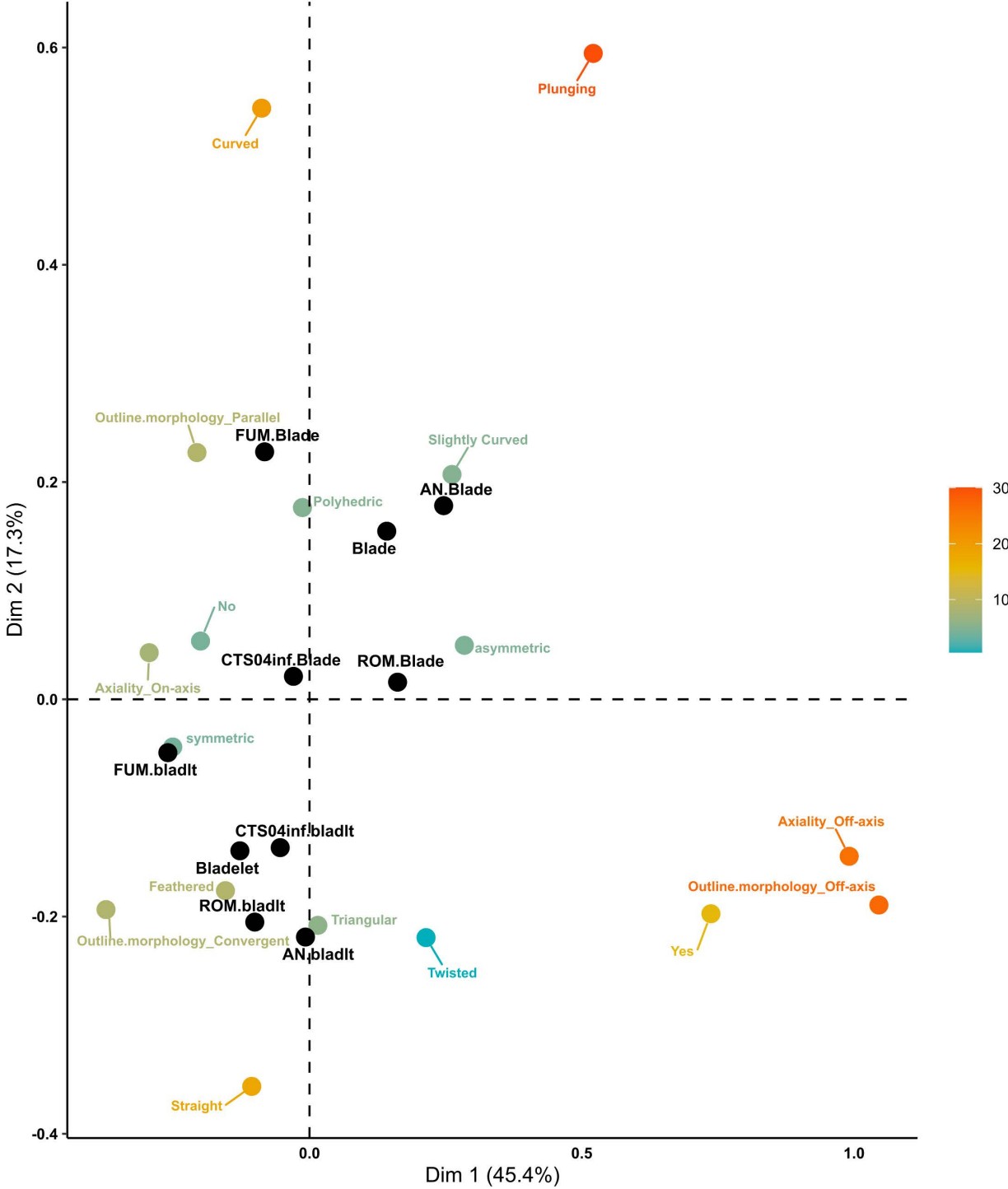

**Fig 10. Convexity domain MCA biplot of EUP assemblages.** The active categories are coloured from light blue to dark orange according to their total contribution to the two dimensions. The supplementary qualitative categories are coloured in black.

attributes, along with the presence of torsion, form a distinct cluster, as indicated by their Euclidean distances. At the opposite end, axial morphologies are associated with the absence of torsion, symmetric cross-section shapes, and convergent outlines. The blade category and AN blades are closest to the off-axis cluster and are positioned nearer to slightly curved profiles and asymmetric cross-sections. FUM bladelets and the bladelet category align more closely with the on-axis group. Overall, the assemblages show some grouping but remain diverse in morphology. Dimension 2 contrasts variables within the same attribute group. Straight versus curved profiles, feathered versus plunging distal ends, and triangular versus polyhedric cross-section shapes are positioned at opposite ends of Dimension 2. Bladelets primarily cluster in the lower left quadrant, associated with feathered terminations, triangular cross-sections, and straight profiles, while blades tend to group in the upper quadrant. ROM and AN blades are characterized by off-axis terminations and asymmetric cross-sections, while FUM and CTS04inf blades display more regular morphologies with curved profiles. The biplot reveals a clear trend: blades from all assemblages predominantly cluster in the upper part of the plot, while bladelets are concentrated in the lower part. This pattern underscores the consistent characteristics within assemblages and highlights the distinct differences between blades and bladelets. It emphasizes the importance of studying them separately. Among the bladelets, FUM bladelets show the highest degree of morphological regularity.

**Retouch domain.** Only blades and bladelets with lateral retouch (n = 572) were analysed, as they represent the most typical tool types of the EUP. The first two dimensions of the MCA explain only 34.4% of the total variance, with Dimension 1 accounting for 25.2% and Dimension 2 for 9.2% (Fig 11). Dimension 1 highlights a contrast between continuous and partial retouch distribution. Retouch positions are largely independent of one another, with inverse retouch being closer to alternate than to direct, as expected. FUM bladelets are associated with alternate, bilateral, and continuous retouch, characteristic of classic Dufour bladelets, while CTS04inf blades, along with ROM and AN bladelets, cluster primarily with partial retouch. Dimension 2 reflects the opposition between direct and inverse retouch Positions. CTS04inf bladelets cluster strongly with the inverse position, while blades, particularly those from ROM and FUM, cluster with the direct position. Additionally, FUM, ROM, and AN's blades tend to cluster with left retouch Locations. Overall, AN and ROM assemblages cluster for both bladelets and blades, while all four sites cluster for blade retouch. This suggests a superregional unity in blade retouch practices. However, bladelet retouch becomes increasingly idiosyncratic moving westward. FUM bladelets as well as CTS04inf bladelets do not form a cluster but instead appear isolated, suggesting a specific retouching style distinct from each of these assemblages when it comes to bladelets. This geographical variability in bladelet retouch has been described in previous studies [42]. The contrast between the unity in blade retouch and the variability in bladelet retouch highlights an intriguing pattern: eastern assemblages, such as AN and ROM, are much more similar to each other despite being attributed to different technocomplexes than to any of the other assemblages studied here.

## Discussion

This study is among the first to compare combined sets of categorical and numerical lithic attributes from unretouched and retouched blanks attributed to different Upper Palaeolithic technocomplexes. It examines their collective variation on a continuous scale using MCA. Below, we will first reflect on the effectiveness of our MCA protocol, based on the attributes grouped into three domains and recorded on both retouched and unretouched blanks, in capturing variation within Upper Palaeolithic laminar productions. Then, we will assess its capacity to detect more subtle variations when focussing on the Early Upper Palaeolithic. Finally, we will look into the interest of MCA to better understand the role of bladelets in the lithic productions of the EUP.

### Capturing variability using MCA within the Upper Palaeolithic

We hypothesise that the studied EUP assemblages would be more similar to one another than to any other Upper Paleolithic assemblages, assuming that technocomplexes close in time, like those grouped in the EUP, would share more

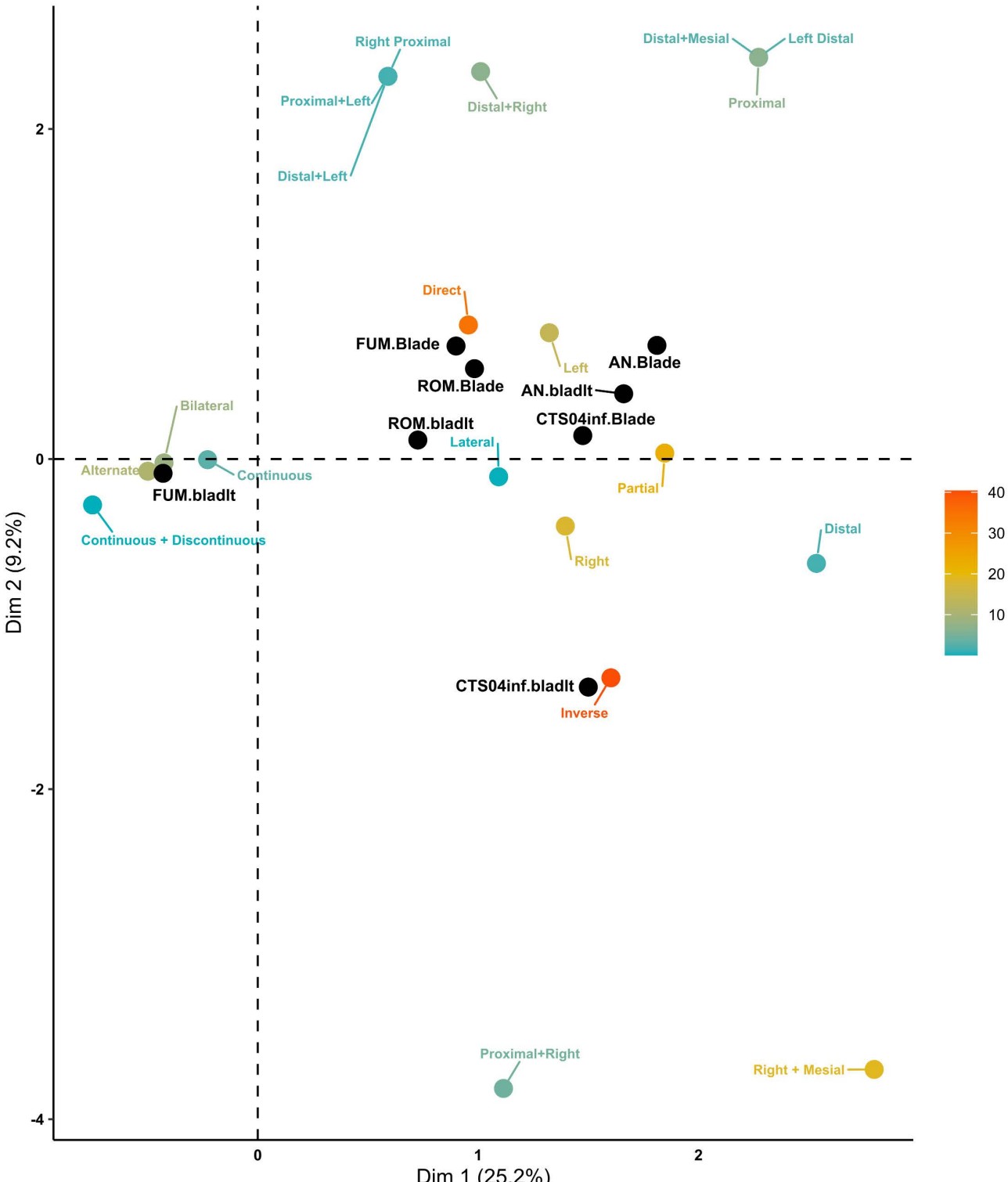

**Fig 11. Retouch domain MCA biplot of EUP assemblages.** The active variables are coloured from light blue to dark orange according to their total contribution to the two dimensions. The supplementary qualitative categories are coloured in black.

technological traits than non-contemporary technocomplexes. Our MCA technological results support this view. Both within the Platform domain (Fig 7) and the Convexity domain (Fig 9), the Solutrean blanks cluster closely together. In contrast, the Protoaurignacian and Southern Ahmarian assemblages are more closely related to each other than either is to the Solutrean. The Protoaurignacian and Southern Ahmarian assemblages are strongly associated with more slender blanks and punctiform or linear platforms, while the Solutrean assemblages are predominantly associated with thicker blanks (relative to their width) and cortical platforms. The Solutrean technocomplex also clusters separately from the EUP, particularly in terms of core convexity shaping and the centrality of knapping on the flaking surface. While the EUP assemblages, especially the Southern Ahmarian, tend to cluster toward off-axis morphologies, torsion, and plunging distal terminations, the Solutrean assemblages are more characterised by on-axis, convergent, and feathered profiles. Furthermore, although no MCA comparison is possible between the EUP and Solutrean assemblages due to the lack of retouch attributes in the Solutrean dataset, the retouch patterns of EUP blades and bladelets are more similar to each other than to those of the Solutrean. In the Solutrean, retouch often transforms blanks into backed, shouldered, stemmed, and winged points (see details in [57]), rather than the lateral retouch typical of the EUP.

## The role of bladelets in the EUP reduction process

One focus of our analysis is whether Palaeolithic archaeologists should assign greater significance to bladelets, which are a rather uncommon blank type before the onset of the Upper Palaeolithic [32]. There is plenty of research on the role of bladelets and its importance in defining the transition to new behaviours and social organisations. For example, Bon [31] and Teyssandier [33] suggested a different role for bladelets in the Protoaurignacian and the following Early Aurignacian: within the first one bladelets are produced on volumetric cores starting as blade cores and ending as large bladelet cores, while in the latter are produced from specialised small cores (carinated cores). The Southern Ahmarian follows the same Protoaurignacian process: the core is eventually reduced and therefore it produces smaller blades, the bladelets [73,116,136]. Though new research suggested that bladelets are not the product of core shrinkage, they are actively sought as target products in Protoaurignacian and Early Southern Ahmarian contexts [50,65,67,71]. They also form the bulk of the most recognisable retouched tool types within the Aurignacian: Dufour bladelets and Font-Yves bladelets [77]. Yet, the 12 mm width threshold traditionally used to differentiate blades from bladelets is an empirical standard that has been widely applied but rarely assessed. Here we will reflect on how and if MCA can help to assess this 12 mm threshold and hence test the strength of this parameter used to distinguish blade from bladelets during the Palaeolithic.

The Convexity domain (Fig 10) shows a neat division between the blades and bladelets of the EUP assemblages. Instead, the Solutrean blades and bladelets are well associated with each other, witnessing a strong similarity across the different sizes (Fig 9). EUP Bladelets align with attributes like feathered, on-axis, convergent, and straight, signifying they mostly belong to target production phases. Instead, blades tend to split between those featuring an asymmetric cross-section and slightly curved profile (AN and ROM) and those showing polyhedric cross-section and parallel outline (CTS04inf and FUM). This confirms the earlier classical analysis. In AN and ROM bladelets tend to be identified as coming from target production phases, while blades split between management and target [50]. In general, bladelets are more regular and elongated than blades [137] but there is some degree of overlapping between blades and bladelets [137,138]. This is mirrored by the Convexity domain results, in fact FUM blades are closer to regular, on-axis morphologies, but nevertheless also to curved profiles. Such emphasis on bladelets within the EUP is also demonstrated by the specific treatment they received in FUM and CTS04inf retouch patterns. Instead, in ROM and AN the retouch pattern is rather unspecialised. Our MCA analysis is then a good indicator that the 12 mm width threshold between blades and bladelets has a heuristic meaning in the EUP technology.

## Capturing variability using MCA within the EUP

Our second hypothesis tested whether there is a consistent association between the assemblages classified in one EUP facies, hence testing the validity of keeping separate the various EUP assemblages according to their traditional

taxonomy. Both the Platform (Fig 8) and Convexity (Fig 10) domains do not show a strong association between assemblages attributed to the Protoaurignacian and isolation of the only Southern Ahmarian assemblage. The Platform domain shows a separation according to geographical gradient, with the ROM and AN assemblages mostly laying in the upper quadrants of the biplot, while FUM and CTS04inf assemblages lie in the lower quadrants. Nevertheless, all the assemblages are mostly clustered around the dimensions' origins, hence showing a low degree of difference. Also, we must notice that plain, punctiform, and linear platforms, despite being a widely accepted terminology in platform description are rather void of meaning if not accompanied by more objective attributes, such as platform measurements. Effectively, the biplot confirms the common knowledge about the EUP assemblage and the exploratory plots: EUP assemblages rarely prepare the striking platforms. To further expand this line of research, it would be useful to include Early Aurignacian assemblages in the analysis, as faceting is commonly witnessed in blade production [31,139,140]. The Convexity domain shows a differentiation according to the blanks category and not a clustering of different technocomplexes. Whether patterns of differentiation within the EUP exist they might be highlighted by attributes reflecting stylistic and functional choices: for example, retouching. The EUP features mostly a various array of laterally retouched bladelets. The most indicative features are the position, localisation, and distribution of retouch identifying them as Dufour bladelets, Font-Yves points, and el-Wad points [77,141–144]. Through our analysis (Fig 11), we show there is a degree of difference between the assemblages in terms of the way blades and bladelets are retouched. FUM is strongly correlated with the classic definition of Dufour bladelets, instead, CTS04inf bladelets show a strong correlation with the inverse position, therefore mostly being of the Dufour sub-type Dufour type. In an earlier comparative study, the prevalence of inverse retouch on Les Cottés Protoaurignacian bladelets and the prevalence of alternate retouch on Grotta di Fumane A1-A2 bladelets was already evident [42]. Instead, blades from FUM, ROM, and CTS04inf are mostly related to the direct position. Also, ROM bladelets mostly correlate with direct retouching, despite the presence of Dufour bladelets in the assemblage [41]. Blades and bladelets from AN correlate more with the direct and partial retouch. Earlier work showed that the el-Wad point is a rather unstandardised type [144] and that retouch does not follow a particular configuration in distribution and localisation [71].

Gennai and colleagues [50] suggested that technological attributes do not support different technocomplexes, but a strong degree of similarity between EUP assemblages in terms of technological behaviour. The present analysis confirms this suggestion, at least for the compared assemblages; in fact, the Al-Ansab AH 1 assemblage fits well within the Protoaurignacian assemblages' variability. Instead, our new results on retouch patterns might shed some light on the regionalisation, or perhaps internal chronological evolution, of the EUP. The current hypotheses of the EUP dispersal generally agree on a rapid east-to-west, movement [1,2,4,35,145,146], although this pattern might be contradicted by evidence of complex networks and mobility strategies even at the onset of the European EUP [41,147,148]. Our findings could be consistent with either a rapid dispersal of human groups carrying a coherent technological set that endured relatively unaltered for millennia, or with ongoing interactions within this geographical and temporal expanse. Future studies involving broader comparative datasets and refined chronological frameworks are essential for addressing these questions.

## Conclusions

With this paper, we would like to affirm the importance of lithic studies and transparent methodologies of investigation to reconstruct past human behaviours and major anthropological events, like one of the *Homo sapiens* dispersals. Lithic technological studies play a pivotal role in complementing genetic research. While DNA studies offer insights into migration patterns, and interbreeding events, lithic analyses provide tangible evidence of cultural transmission, adaptation, and ecological interactions [26,46,58]. For instance, the shared technological traits between the Southern Ahmarian and Protoaurignacian may corroborate the hypothesis of shared genetic ties between Europe and SW Asia at the time. It also reflects aDNA evidence showing distinct genetic traits during the Aurignacian [26].

The reproducible methodology employed in this study, including the open sharing of datasets and analytical workflows, sets the foundation for future interdisciplinary research on the early Upper Palaeolithic. It is part of a broader movement in Palaeolithic archaeology aimed at improving reproducibility and data-sharing [47,62,126,149]. Our analysis addressed key topics of debate for the reconstruction of Early Upper Palaeolithic behaviours, such as the similarity between technocomplexes and the role of bladelets within the reduction process. Whether this is the results of phyletic evolution, exchanges or independent developments would require more integration of chronological, cultural and genetic data. As lithic technologists, we notice that both technocomplexes show a similar attitude towards bladelet production and that bladelets are seemingly more standardised than blades. This standardisation is emphasised by the specific retouch patterns of bladelets in the Les Cottés 04 inférieur and Grotta di Fumane A1-A2 assemblages. Furthermore, these retouch patterns are possibly indicating differences within the analysed EUP assemblages. These differences might be related to functionality, but also by chronological or geo-ecological dynamics.

As we continue to refine and expand the technological and genetic evidence, we move closer to constructing a holistic narrative of the transitions within the Upper Palaeolithic and the spread of modern humans across Eurasia.

## Supporting information

**S1 File. A text with a detailed summary of the Aurignacian and Ahmarian research history.**
(PDF)

**S2 File. An excel file compiling radiocarbon dates of Early Upper Palaeolithic sites.**
(XLSX)

**S3 File. An excel file compiling coordinates of Early Upper Palaeolithic sites.**
(XLSX)

**S1 Fig. A zipped folder with all supporting information figures.**
(ZIP)

## Acknowledgments

Research at Al-Ansab 1 and Românești-Dumbrăvița I were carried out within the framework of the SFB 806 "Our Way to Europe". Al-Ansab 1 was excavated in collaboration with the Department of Antiquities (Amman/Jordan) the University of Jordan/ Amman participation. Românești-Dumbrăvița I 2016 excavation campaign was carried out in collaboration with the the Museum of Banat and the 2018–2019 excavation campaigns in collaboration with the Vasile Pârvan Institute of Archaeology of the Romanian Science Academy in Bucharest. Jacopo Gennai acknowledges Alexandru Ciornei, Wei Chu, and Adrian Dobos for their work at Românești-Dumbrăvița, and Florian Sauer, Marcel Schemmel, and Jonathan Schoenberg for their research at Al-Ansab AH 1. Research at Fumane is coordinated by the University of Ferrara in the framework of a project supported by the Ministry of Culture – Veneto Archaeological Superintendency, public institutions (Lessinia Regional Natural Park, Fumane Municipality, BimAdige), and private associations and companies. Excavations at Les Cottés were allowed by the cave owner, J. Bachelier and C.-H. Bachelier, and excavation permits were issued by the Service regional de l'Archéologie d'Aquitaine. We thank J. and C.-H. Bachelier for allowing us to curate the collection excavated by us on their land.

## Author contributions

**Conceptualization:** Jacopo Gennai, Armando Falcucci, Marie Soressi.

**Data curation:** Jacopo Gennai, Armando Falcucci.

**Formal analysis:** Jacopo Gennai, Armando Falcucci, Vincent Niochet.

**Funding acquisition:** Armando Falcucci, Marco Peresani, Jürgen Richter, Marie Soressi.

**Investigation:** Jacopo Gennai, Armando Falcucci, Vincent Niochet.

**Methodology:** Jacopo Gennai, Armando Falcucci.

**Project administration:** Jacopo Gennai.

**Resources:** Marco Peresani, Jürgen Richter, Marie Soressi.

**Validation:** Jacopo Gennai, Armando Falcucci.

**Visualization:** Jacopo Gennai.

**Writing – original draft:** Jacopo Gennai, Armando Falcucci, Vincent Niochet, Marco Peresani, Jürgen Richter, Marie Soressi.

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
