## [Decision Letter · Decision Letter 0]

29 Apr 2025

Dear Dr. Falcucci,

Thank you for submitting your manuscript to PLOS ONE. After careful consideration, we feel that it has merit but does not fully meet PLOS ONE’s publication criteria as it currently stands. Therefore, we invite you to submit a revised version of the manuscript that addresses the points raised during the review process.

We look forward to receiving your revised manuscript.

Kind regards,

Enza Elena Spinapolice, Ph.D

Academic Editor

PLOS ONE

Journal Requirements:

2. In your manuscript, please provide additional information regarding the specimens used in your study. Ensure that you have reported human remain specimen numbers and complete repository information, including museum name and geographic location.

For more information on PLOS ONE's requirements for paleontology and archeology research, see https://journals.plos.org/plosone/s/submission-guidelines#loc-paleontology-and-archaeology-research .

3. Please include a complete copy of PLOS’ questionnaire on inclusivity in global research in your revised manuscript. Our policy for research in this area aims to improve transparency in the reporting of research performed outside of researchers’ own country or community. The policy applies to researchers who have travelled to a different country to conduct research, research with Indigenous populations or their lands, and research on cultural artefacts. The questionnaire can also be requested at the journal’s discretion for any other submissions, even if these conditions are not met.  Please find more information on the policy and a link to download a blank copy of the questionnaire here: https://journals.plos.org/plosone/s/best-practices-in-research-reporting. Please upload a completed version of your questionnaire as Supporting Information when you resubmit your manuscript.

4. Please include a complete copy of PLOS’ questionnaire on inclusivity in global research in your revised manuscript. Our policy for research in this area aims to improve transparency in the reporting of research performed outside of researchers’ own country or community. The policy applies to researchers who have travelled to a different country to conduct research, research with Indigenous populations or their lands, and research on cultural artefacts. The questionnaire can also be requested at the journal’s discretion for any other submissions, even if these conditions are not met.  Please find more information on the policy and a link to download a blank copy of the questionnaire here: https://journals.plos.org/plosone/s/best-practices-in-research-reporting. Please upload a completed version of your questionnaire as Supporting Information when you resubmit your manuscript.

5. Please include captions for your Supporting Information files at the end of your manuscript, and update any in-text citations to match accordingly. Please see our Supporting Information guidelines for more information: http://journals.plos.org/plosone/s/supporting-information .

6. We note that there is identifying data in the Supporting Information file <SI-File 2_Dates_EUP.xlsx and SI-File 3_Sites_Aurignacian.xlsx>. Due to the inclusion of these potentially identifying data, we have removed this file from your file inventory. Prior to sharing human research participant data, authors should consult with an ethics committee to ensure data are shared in accordance with participant consent and all applicable local laws.

-Location data

Please remove or anonymize all personal information (Name, ID,ensure that the data shared are in accordance with participant consent, and re-upload a fully anonymized data set. Please note that spreadsheet columns with personal information must be removed and not hidden as all hidden columns will appear in the published file.

Reviewers' comments:

Reviewer's Responses to Questions

**Comments to the Author**

1. Is the manuscript technically sound, and do the data support the conclusions?

Reviewer #1: Yes

Reviewer #2: Yes

2. Has the statistical analysis been performed appropriately and rigorously?

Reviewer #1: Yes

Reviewer #2: Yes

3. Have the authors made all data underlying the findings in their manuscript fully available?

Reviewer #1: Yes

Reviewer #2: Yes

4. Is the manuscript presented in an intelligible fashion and written in standard English?

Reviewer #1: Yes

Reviewer #2: Yes

Reviewer #1: This is an interesting manuscript that – after revision – should be published.

The methods used are sound and well justified. The authors do a careful analysis and describe their analytical steps in detail.

Their findings fit well with what has been suggested before – similarity of Southern Ahmarian and Proto-Aurignacian across Near East and Europe. What is interesting on their study is that they treat retouch location, convexity management and platform preparation separately and investigate variation within those domains.

However, I also have some critical remarks to the analysis or the presentation and discussion of the results, for these please see below.

Now some more specific comments:

With regard of the “domains of lithic technology” – This has been introduced as a concept by Tostevin (book 2012) and then subsequently used by Nigst (book 2012) in their analyses of EUP assemblages. However, Tostevin and Nigst used more domains. It would be interesting to know why the authors chose to use only three of the domains? Why those three and not the others? – Tostevin and subsequently also Nigst used the domains to test for similarities/differences in generalized knapping behaviour, and specifically to test for horizontal cultural transmission. Tostevin and Nigst explicitly avoided to single out specific blank types (bladelets or blades). The authors mention that they use three “key” domains. Why are the three domains used thought to be “key” domains and not the others? – I am not saying that I disagree with the authors, but in my opinion clearly stating the reasons for their selection would make their argumentation stronger and their analystical choices more transparent.

Results section – Exploratory plots (lines 584-597)

I am missing references to the specific exploratory plots. Please add figure captions to all SI plots.

Results section – Metrical data of …. (lines 598-613)

This section need significant reworking and updating.

Line 599: “…histograms of width…” width of what…?

Line 601: The authors refer to “density distributions” but those are not shown on any of the figures. Where are they? And, if included, hwo are the density distributions calculated? Which algoritms used for density estimation (kernel density?)?

Line 603: The mode is not the bin with highest density but highest frequency of observations.

Line 604-605: the mode for ROM is not 9 mm but 11 mm; the mode for FUM is not 10.5 mm but 8 mm, the mode for CTS04inf is not 10.5 mm but 11 mm (if I read their histograms in Fig 6 correctly).

Line 606: what are “combined samples” – are these blades and bladelets? – If so, make it obvious.

Line 611-613: Which test was used when looking for “meaningful” variations between the assemblages of the three used technocomplexes? How is meaningful variation defined?

Line 614/Fig 6 caption: what are “available” widths?

Line 620: What are “entry artefacts”?

Lines 705-706: “most typical tool types of the EUP” – What are “typical tool types”? Are we not projecting our assumption of the importance of retouched bladelets and blades on EUP humans? – Most tool types are already problematic in their definition – if we then start to distinguish “typical” tool types from (I assume) non-typical tool types…. – I think what is needed here is a clear justification of why you look at retouched bladelets and retoched blades and why not at others. Has this something to do with how frequent certain tool types occur?

Line 738: “look into the interest of MCA analysis” – What is the “interest”?

Also: MCA is already the abbreviation of “Multiple Correspondence Analysis” – hence, the “analysis” after MCA should be deleted.

Line 793: “Pseudo-Dufour bladelets” are not a standard terminology – please define clearly with references and/or state what other retouched bladelets you consider as Pseudo-Dufour.

Line 816-817: this needs re-phrasing. Both Proto-Aurignacian and Early Aurigncian bladelets are the result of core reduction. Just the cores are very different. I suggest to re-phrase as follows: “within the first one bladelets are produced on volumetric cores starting as blade cores and ending as large bladelet cores, while in the latter are produced from specialised small cores (carinated cores)”.

Line 845: “technological” should be replaced with “Lithic technological”.

Line 852-853: “…sets a precedent for future interdisciplinary research”. This suggests that the study is interdisciplinary. I am not sure we can say that the research reported in the MS under review is interdisciplinary. It is a nice study using quantitative methods, but this does not make it interdisciplinary!

Neither Kostenki 14 burial nor the Muierii skull are related to the Aurignacian – while both fall in the range of Aurignacian sensu lato, they are not stratigraphically associated with Aurignacian lithics or other material culture remains. Hence, refs in line 58 need updating or a different phrasing for human remains dated the Aurignacian time-window but not directly associated with Aurignacian objects/occupations.

The Referencing throughout the MS needs some updating:

*) Ignores work by Ofer Bar-Yosef and Jiri Svoboda (Brno) on Bohunician as well as that of Gilbert Tostevin (Minnesota) and Philip Nigts (Cambridge/Vienna) – especially in lines 43-45 but also in lines 49-50. This is especially strange as the authors use a quantification approach that is somewhat similar to the work by Tostevin and Nigst – at least more similar than the Demidenko & Skrdla and Skrdla work…

*) Line 57: One shpuld reference here to Paul Mellars but esp also to Will Davies (Southampton).

*) Line 328: In what ways are refs no 28, 67, 74, 107 and 116 to 120 relevant here? – None of these papers says anything about CTS…

*) Line 331: In what ways are refs no 120 and 121 relevant here? – None of the two papers says anything about CTS…

*) When discussing the EUP in general and the Aurignacian (Proto-Aurigncain as well as Ealry Aurignacian) the MS currently ignores the work of William Davies (Southampton) – someone who has argued for a long time for a rapid dispersal of modern humans into and within Europe. See, e.g., lines 802-804.

Throughout the MS the following term need updating: “mesial” is not used in English lithic terminology (except if French/Italian/Spanish native speakers are the authors). Please replace in text, figure and table captions, tables and all SI information including SI tables the term “mesial” (and variants like “mesidistal” etc.) with “medial”. See, e.g., Shea, Stone tools in the Paleolithic and Neolithic Near East: A guide, Cambridge University Press, 2007, Fig 2.7

Desciption of US04-inf of Les Cottes in lines 322-326 includes information on the site that should be moved to the description of the site (lines 222-248).

I suggest using “southern East European Plain” instead of “southern Russian Plain” (line 126) – both has been used in past (https://en.wikipedia.org/wiki/East_European_Plain) and “Russian Plain” is not per se wrong, but due to Russian invasion of Ukraine I ask the authors to re-consider the term “Russian Plain” here.

I like that the authors include all data as supporting information, which is becoming more frequent but is still rare.

I would also suggest to the authors that they include their R code as supporting information or deposit it on zenodo.org or osf – this would increase reproducibility of their study.

Reviewer #2: In the article “Tracking the emergence of the Upper Palaeolithic with Multiple Correspondence

Analysis of Protoaurignacian and southern Ahmarian lithic assemblages”, Gennai and colleagues present a detailed and innovative study of four Early Upper Palaeolithic sites in Eurasia, critically examining the notion that the taxonomic division of lithic industries into discrete entities—so-called technocomplexes—may in fact obscure a deeper underlying variability, thereby leading to potential misinterpretations of Homo sapiens’ dispersal dynamics. Through the application of multivariate statistics, the authors conduct a comparative analysis of four lithic assemblages to evaluate both their technological proximity or divergence (i.e., the existence and definition of a technocomplex) and their distinctiveness in relation to broader Upper Palaeolithic cultural traditions.

The article is well written, intelligently conceived, and excellently structured. Crucially, it provides a much-needed comparative analysis that engages seriously with the issue of comparability across lithic assemblages,, moving beyond the commonly accepted emphasis on metric attributes—those most easily compared, as highlighted in the literature—and making the extra collaborative effort to standardise an extensive dataset into a single, open-access database. This is a non minor achievement in the field of lithic analysis and contributes meaningfully to a growing body of research that underscores the importance of data sharing, transparency, replicability, and the broader reusability of datasets to advance our collective goal of reconstructin past human behaviours

Therefore, in light of the methodological rigour of the approach, the clarity and structure of the article, the significance of the results obtained, and the collaborative effort of the authors, I fully support the publication of this paper.

I would, however, like to offer a few suggestions which I believe could further improve the readability of the manuscript and enhance the clarity of the analysis:

1. Clarify analytical categories

While the explanation of the analysed contexts and technological strategies is extensive and detailed, it would be beneficial to state clearly from the beginning (abstract) which artefact categories were considered. The deliberate exclusion of cores, for example, should be explicitly mentioned and briefly justified.

2. Terminological clarification

The manuscript makes reference to "blanks", yet the analysis also incorporates the retouch domain as a meaningful dimension of variability. It would be helpful to define what is meant by "blank" in this context and explain its relation to the domains used in the study.

3. lines 40-42: the sentence discussing complex bio-cultural dynamics would be strengthened by adding a relevant reference to support the claims regarding large-scale synchronicity and regional developments.

4. Line 121: A reference should be added to explain the use of a 12 mm threshold to distinguish blades from bladelets, as this is a well-debated convention in the literature.

5. From Line 462: Please consider to clarify the classification system from which platform types were reduced to the current categories. Also, the decision to merge dihedral and facetted platforms should be justified—perhaps due to their scarcity in certain assemblages?

6. Since the attributes related to retouch are well detailed, it would be helpful to elaborate on how assemblages were grouped for comparative purposes. A brief explanation of the rationale behind the comparative units would aid interpretation.

7. Consider including a table summarising which variables compose each of the meaningful domains (platform preparation, convexity management, retouch, etc.). This would greatly enhance readability and help the reader follow the analytical structure.

8. Moreover, While the manuscript is generally well written, there are some minor issues with English grammar, particularly the omission of prepositions in certain phrases. A careful language review would be beneficial to ensure clarity and consistency throughout the text.

These are intended as constructive suggestions to further improve the clarity and accessibility of what is already a rigorous and valuable contribution to the study of the Early Upper Palaeolithic.

**Do you want your identity to be public for this peer review?** For information about this choice, including consent withdrawal, please see our Privacy Policy

Reviewer #1: No

Reviewer #2: **Yes: ** Marianna Fusco

---

## [Author Response · Author response to Decision Letter 1]

13 Jun 2025

Dear anonymous reviewer #1, Dr. Fusco, and Prof. Spinapolice,

We are glad about your overall appreciation of our manuscript and your comments leading to the manuscript's improvement and final publication. Please find below a detailed answer to your comments. We marked the comments with #sequential number and we give a reply (Reply:) underneath.

Reviewer #1: This is an interesting manuscript that – after revision – should be published.

The methods used are sound and well justified. The authors do a careful analysis and describe their analytical steps in detail. Their findings fit well with what has been suggested before – similarity of Southern Ahmarian and Proto-Aurignacian across Near East and Europe. What is interesting on their study is that they treat retouch location, convexity management and platform preparation separately and investigate variation within those domains. However, I also have some critical remarks to the analysis or the presentation and discussion of the results, for these please see below.

Now some more specific comments:

#1 comment: With regard of the “domains of lithic technology” – This has been introduced as a concept by Tostevin (book 2012) and then subsequently used by Nigst (book 2012) in their analyses of EUP assemblages. However, Tostevin and Nigst used more domains. It would be interesting to know why the authors chose to use only three of the domains? Why those three and not the others? – Tostevin and subsequently also Nigst used the domains to test for similarities/differences in generalized knapping behaviour, and specifically to test for horizontal cultural transmission. Tostevin and Nigst explicitly avoided to single out specific blank types (bladelets or blades). The authors mention that they use three “key” domains. Why are the three domains used thought to be “key” domains and not the others? – I am not saying that I disagree with the authors, but in my opinion clearly stating the reasons for their selection would make their argumentation stronger and their analytical choices more transparent.

Reply:

We thank Reviewer 1 for their comments. We agree that “key” domains were an unclear wording and have now removed the word “key”. We did not use the domains defined by Tostevin based on cores because we decided not to use cores to avoid the pitfalls of the end of the reduction stages. We used the domains based on blanks and retouched artefacts descriptions and removed the attributes, like EPA angle, that can have major instrumental and observer errors. Now we have acknowledged more profusely Nigst and Tostevin’s work. Nevertheless, we would like to point out that our domains are an independent convergence, not a straight use of Tostevin’s ones. Also, the focus on singling out blades and bladelets is due to the specific questions we had about the Early Upper Palaeolithic, a period when technocomplexes produce significantly more bladelets than before and because of the role assigned to Aurignacian bladelet-making by European authors

#2 comment: Results section –Exploratory plots (lines 584-597). I am missing references to the specific exploratory plots. Please add figure captions to all SI plots.

Reply: we added the requested captions

#3 comment: Results section – Metrical data of …. (lines 598-613). This section need significant reworking and updating.

Line 599: “…histograms of width…” width of what…?

Reply: We are referring to the analysed blanks, unretouched blades, bladelets and the combined unretouched blades and bladelets sample.

#4 comment: Line 601: The authors refer to “density distributions” but those are not shown on any of the figures. Where are they? And, if included, hwo are the density distributions calculated? Which algoritms used for density estimation (kernel density?)?

Reply: We agree: “density distributions” is not the proper term and therefore we removed it.

#5 comment: Line 603: The mode is not the bin with highest density but highest frequency of observations.

Reply: Thank you for the correction.

#6 comment: Line 604-605: the mode for ROM is not 9 mm but 11 mm; the mode for FUM is not 10.5 mm but 8 mm, the mode for CTS04inf is not 10.5 mm but 11 mm (if I read their histograms in Fig 6 correctly).

Reply: Thank you for pointing this out. We agree—the initial mode values were inaccurately reported due to a miscalculation. We have now recalculated the exact frequency counts in 1 mm bins for each site, and the corrected mode values are as follows: ROM peaks in the 11–12 mm bin (42 counts), not at 9 mm; FUM peaks in the 9–10 mm bin (51 counts), not at 10.5 mm; and CTS04inf peaks in the 10–11 mm bin (39 counts), not at 10.5 mm. It is also worth noting that ROM shows a nearly equal secondary peak in the 8–9 mm bin (41 counts), indicating a bimodal distribution. These corrections have been incorporated into the revised version of the manuscript.

#7 comment: Line 606: what are “combined samples” – are these blades and bladelets? – If so, make it obvious.

Reply: your interpretation is correct, now we made it more explicit.

#8 comment: Line 611-613: Which test was used when looking for “meaningful” variations between the assemblages of the three used technocomplexes? How is meaningful variation defined?

Reply: No test has been used, that is why we used “meaningful” over “significant”, if you prefer we are switching to “visible”.

#9 comment Line 614/Fig 6 caption: what are “available” widths?

Reply: we changed into “blades and bladelets” so now it is more explicit.

#10 comment: Line 620: What are “entry artefacts”?

Reply: it is a slip, we meant database entries, now we corrected

#11 comment: Lines 705-706: “most typical tool types of the EUP” – What are “typical tool types”? Are we not projecting our assumption of the importance of retouched bladelets and blades on EUP humans? – Most tool types are already problematic in their definition – if we then start to distinguish “typical” tool types from (I assume) non-typical tool types…. – I think what is needed here is a clear justification of why you look at retouched bladelets and retoched blades and why not at others. Has this something to do with how frequent certain tool types occur?

Reply: your interpretation is right. We do not assume any emic importance of retouched bladelets to EUP human groups. Nevertheless, some tool types according to the existing literature, such as Dufour bladelets and other marginally retouched bladelets, are indeed more frequent, and therefore typical, in EUP archaeological contexts, As opposed to simple endscrapers or simple burins that are rather common throughout the whole Upper Palaeolithic. Hence, we believe that marginally retouched bladelets do hold at least an archaeological significance into interpreting EUP contexts.

#12 comment: Line 738: “look into the interest of MCA analysis” – What is the “interest”? Also: MCA is already the abbreviation of “Multiple Correspondence Analysis” – hence, the “analysis” after MCA should be deleted.

Reply: corrected

#13 comment: Line 793: “Pseudo-Dufour bladelets” are not a standard terminology – please define clearly with references and/or state what other retouched bladelets you consider as Pseudo-Dufour.

Reply: Thank you for your suggestion. We agree and we have removed these terms and adjusted the text.

#14 comment: Line 816-817: this needs re-phrasing. Both Proto-Aurignacian and Early Aurigncian bladelets are the result of core reduction. Just the cores are very different. I suggest to re-phrase as follows: “within the first one bladelets are produced on volumetric cores starting as blade cores and ending as large bladelet cores, while in the latter are produced from specialised small cores (carinated cores)”.

Reply: we accepted your suggestion.

#15 comment: Line 845: “technological” should be replaced with “Lithic technological”.

#16 comment: Line 852-853: “…sets a precedent for future interdisciplinary research”. This suggests that the study is interdisciplinary. I am not sure we can say that the research reported in the MS under review is interdisciplinary. It is a nice study using quantitative methods, but this does not make it interdisciplinary!

Reply: We understand how this could have been misleading therefore we revised the sentence as such: .

“The reproducible methodology employed in this study, including the open sharing of datasets and analytical workflows, sets the foundation for future interdisciplinary research on the early Upper Palaeolithic.”

#17 comment: Neither Kostenki 14 burial nor the Muierii skull are related to the Aurignacian – while both fall in the range of Aurignacian sensu lato, they are not stratigraphically associated with Aurignacian lithics or other material culture remains. Hence, refs in line 58 need updating or a different phrasing for human remains dated the Aurignacian time-window but not directly associated with Aurignacian objects/occupations.

Reply: You are right, and we amended it following your suggestions

#18 comment: The Referencing throughout the MS needs some updating:

*) Ignores work by Ofer Bar-Yosef and Jiri Svoboda (Brno) on Bohunician as well as that of Gilbert Tostevin (Minnesota) and Philip Nigts (Cambridge/Vienna) – especially in lines 43-45 but also in lines 49-50. This is especially strange as the authors use a quantification approach that is somewhat similar to the work by Tostevin and Nigst – at least more similar than the Demidenko & Skrdla and Skrdla work…

Reply: Thank you for your suggestion. We acknowledge the significance of the work by Bar-Yosef & Svoboda, as well as Tostevin and Nigst, in advancing our understanding of the Bohunician and the Initial Upper Palaeolithic more broadly. However, our paper does not focus specifically on the IUP or the detailed technological traits of the Bohunician. For this reason, we opted to reference sources that synthesise regional patterns and are more accessible to a wider readership.

*) Line 57: One shpuld reference here to Paul Mellars but esp also to Will Davies (Southampton).

Reply: we added the requested references

*) Line 328: In what ways are refs no 28, 67, 74, 107 and 116 to 120 relevant here? – None of these papers says anything about CTS…

*) Line 331: In what ways are refs no 120 and 121 relevant here? – None of the two papers says anything about CTS…

Reply: we updated the paragraphs and referencing as suggested.

*) When discussing the EUP in general and the Aurignacian (Proto-Aurigncain as well as Ealry Aurignacian) the MS currently ignores the work of William Davies (Southampton) – someone who has argued for a long time for a rapid dispersal of modern humans into and within Europe. See, e.g., lines 802-804.

#19 comment: Throughout the MS the following term need updating: “mesial” is not used in English lithic terminology (except if French/Italian/Spanish native speakers are the authors). Please replace in text, figure and table captions, tables and all SI information including SI tables the term “mesial” (and variants like “mesidistal” etc.) with “medial”. See, e.g., Shea, Stone tools in the Paleolithic and Neolithic Near East: A guide, Cambridge University Press, 2007, Fig 2.7

Reply: we have amended it.

#20 comment: Desciption of US04-inf of Les Cottes in lines 322-326 includes information on the site that should be moved to the description of the site (lines 222-248).

Reply: we amended the paragraph as suggested.

#21 comment: I suggest using “southern East European Plain” instead of “southern Russian Plain” (line 126) – both has been used in past (https://en.wikipedia.org/wiki/East_European_Plain) and “Russian Plain” is not per se wrong, but due to Russian invasion of Ukraine I ask the authors to re-consider the term “Russian Plain” here.

Reply: we agree with your suggestion.

#22 comment: I like that the authors include all data as supporting information, which is becoming more frequent but is still rare.

I would also suggest to the authors that they include their R code as supporting information or deposit it on zenodo.org or osf – this would increase reproducibility of their study.

Reply: We do share our files, raw and derived databases, and Rmarkdown files with codes in the Open Access Zenodo folder we have included in the Supporting Information

Reviewer #2:

In the article “Tracking the emergence of the Upper Palaeolithic with Multiple Correspondence Analysis of Protoaurignacian and southern Ahmarian lithic assemblages”, Gennai and colleagues present a detailed and innovative study of four Early Upper Palaeolithic sites in Eurasia, critically examining the notion that the taxonomic division of lithic industries into discrete entities—so-called technocomplexes—may in fact obscure a deeper underlying variability, thereby leading to potential misinterpretations of Homo sapiens’ dispersal dynamics. Through the application of multivariate statistics, the authors conduct a comparative analysis of four lithic assemblages to evaluate both their technological proximity or divergence (i.e., the existence and definition of a technocomplex) and their distinctiveness in relation to broader Upper Palaeolithic cultural traditions.

The article is well written, intelligently conceived, and excellently structured. Crucially, it provides a much-needed comparative analysis that engages seriously with the issue of comparability across lithic assemblages,, moving beyond the commonly accepted emphasis on metric attributes—those most easily compared, as highlighted in the literature—and making the extra collaborative effort to standardise an extensive dataset into a single, open-access database. This is a non minor achievement in the field of lithic analysis and contributes meaningfully to a growing body of research that underscores the importance of data sharing, transparency, replicability, and the broader reusability of datasets to advance our collective goal of reconstructin past human behaviours

Therefore, in light of the methodological rigour of the approach, the clarity and structure of the article, the significance of the results obtained, and the collaborative effort of the authors, I fully support the publication of this paper.

I would, however, like to offer a few suggestions which I believe could further improve the readability of the manuscript and enhance the clarity of the analysis:

#1 comment: Clarify analytical categories. While the explanation of the analysed contexts and technological strategies is extensive and detailed, it would be beneficial to state clearly from the beginning (abstract) which artefact categories were considered. The deliberate exclusion of cores, for example, should be explicitly mentioned and briefly justified.

#2 comment. Terminological clarification. The manuscript makes reference to "blanks", yet the analysis also incorporates the retouch domain as a meaningful dimension of variability. It would be helpful to define what is meant by "blank" in this context and explain its relation to the domains used in the study.

#3 comment: lines 40-42: the sentence discussing complex bio-cultural dynamics would be strengthened by adding a relevant reference to support the claims regarding large-scale synchronicity and regional developments.

Reply: We thank Dr. Fusco. We added references as asked

#4 comment: Line 121: A reference should be added to explain the use of a 12 mm threshold to distinguish blades from bladelets, as this is a well-debated convention in the literature.

Reply: We agree. We chose the threshold as it has been widely applied within recent lithic technological analysis of EUP European and Near Eastern assemblages. It is also justified by the empirical fact that retouched tools, such as retouched blades, are often under this threshold.

#5 comment: From Line 462: Please consider to clarify the classification system from which platform types were reduced to the current categories. Also, the decision to merge dihedral and facetted platforms should be justified—perhaps due to their scarcity in certain assemblages?

Reply: Actually, dihedral platforms are not common in the EUP. Most of the debate, if there is even one, revolves around the fact that Early Aurignaci

---

## [Decision Letter · Decision Letter 1]

15 Aug 2025

Tracking the emergence of the Upper Palaeolithic in western Asia and Europe: a Multiple Correspondence Analysis of Protoaurignacian and southern Ahmarian lithics

PONE-D-25-07148R1

Dear Dr. Falcucci

We’re pleased to inform you that your manuscript has been judged scientifically suitable for publication and will be formally accepted for publication once it meets all outstanding technical requirements.

Kind regards,

Enza Elena Spinapolice, Ph.D

Academic Editor

PLOS ONE

Additional Editor Comments (optional):

Reviewers' comments:

Reviewer's Responses to Questions

**Comments to the Author**

Reviewer #2: All comments have been addressed

2. Is the manuscript technically sound, and do the data support the conclusions?

Reviewer #2: Yes

3. Has the statistical analysis been performed appropriately and rigorously?

Reviewer #2: Yes

4. Have the authors made all data underlying the findings in their manuscript fully available?

Reviewer #2: Yes

5. Is the manuscript presented in an intelligible fashion and written in standard English?

Reviewer #2: Yes

Reviewer #2: The authors have addressed all the requested revisions, and the manuscript has been substantially improved. I am satisfied with the changes made, and I now support the publication of the paper in its current form.

**Do you want your identity to be public for this peer review?** For information about this choice, including consent withdrawal, please see our Privacy Policy

Reviewer #2: **Yes: ** Marianna Fusco

---

## [Editor Report · Acceptance letter]

PONE-D-25-07148R1

PLOS ONE

Dear Dr. Falcucci,

I'm pleased to inform you that your manuscript has been deemed suitable for publication in PLOS ONE. Congratulations! Your manuscript is now being handed over to our production team.

Kind regards,

on behalf of

Dr. Enza Elena Spinapolice

Academic Editor

PLOS ONE